# Cerebellar output shapes cortical preparatory activity during motor adaptation

Sharon Israely [1,6], Hugo Ninou [1,2,3,6], Ori Rajchert [4], Lee Elmaleh[1], Ran Harel [5], Firas Mawase [4], Jonathan Kadmon [1] ✉ & Yifat Prut [1] ✉

The cerebellum plays a key role in motor adaptation by driving trial-to-trial recalibration of movements based on previous errors. In primates, cortical correlates of adaptation are encoded already in the pre-movement motor plan, but these early cortical signals could be driven by a cerebellar-to-cortical information flow or evolve independently through intracortical mechanisms. To address this question, we trained female macaque monkeys to reach against a viscous force field (FF) while blocking cerebellar outflow. The cerebellar block led to impaired FF adaptation and a compensatory, re-aiming-like shift in motor cortical preparatory activity. In the null-field conditions, the cerebellar block altered neural preparatory activity by increasing task-representation dimensionality and impeding generalization. A computational model indicated that low-dimensional (cerebellar-like) feedback is sufficient to replicate these findings. We conclude that cerebellar signals carry task structure information that constrains the dimensionality of the cortical preparatory manifold and promotes generalization. In the absence of these signals, cortical mechanisms are harnessed to partially restore adaptation.

Motor adaptation is a remarkable mechanism utilized by all living beings to adjust to changes in both the external environment and the internal physiological state, an ability which supports routine behavior and long-term survival[1,2]. Numerous studies have shown that motor adaptation is achieved by tuning an internal model of the limbs[3,4]. In vertebrates, the cerebellum is considered to be the site where internal models are stored and is therefore responsible for the trial-by-trial recalibration of motor commands[5–8]. This assumption is supported by considerable data from animal models[9–11], imaging studies[12], and observations of cerebellar patients[13–18] who have difficulties adapting to novel environments. However, the mechanisms by which the cerebellar-based internal model gains access to motor output during sensorimotor adaptation remain unclear.

In the upper limb system of humans and nonhuman primates, cerebellar signals are predominantly relayed through the motor thalamus to the motor cortex[19], suggesting that adaptive motor behavior evolves through cerebellar-to-cortical interactions. Consistent with this view, large parts of the sensorimotor network, including the motor, premotor and somatosensory cortices[20–25] have been shown to correlate with motor learning as early as the planning phase of movements[22,26,27]. These early signals may have evolved through intracortical mechanisms to integrate with cerebellar signals when movements begin. Alternatively, adaptation-related cortical signals might

[1]The Edmond and Lily Safra Center For Brain Sciences, The Hebrew University, Jerusalem, Israel. [2]Département D'Etudes Cognitives, Ecole Normale Supérieure, Laboratoire de Neurosciences Cognitives et Computationnelles, INSERM U960, PSL University, Paris, France. [3]Laboratoire de Physique de l'Ecole Normale Superieure, Ecole Normale Supérieure, PSL University, Paris, France. [4]Faculty of Biomedical Engineering, Technion - Israel Institute of Technology, Haifa, Israel. [5]Department of Neurosurgery, Sheba Medical Center, Tel Aviv, Israel. [6]These authors contributed equally: Sharon Israely, Hugo Ninou. ✉e-mail: jonathan.kadmon@mail.huji.ac.il; yifat.prut@mail.huji.ac.il

reflect pre-movement cerebellar-to-cortical interactions that occur during this time[28–30].

Here, we aimed to delineate the cerebellar contribution to motor adaptation by training monkeys to reach against a velocity-dependent force field while cerebellar output was reversibly blocked[31]. Blocking cerebellar signals resulted in a significantly impaired adaptation, consistent with observation in cerebellar patients[32,33]. The impaired motor behavior was preceded by changes in pre-movement neural activity, characterized by a rotational shift in neural state, and a significant increase in the dimensionality of activity compared to control conditions. Finally, under these circumstances, across-target generalization was impaired at the neural and behavioral levels. A computational model explained these findings by showing that low-dimensional (cerebellar-like) feedback is sufficient to quench the dimensionality of the cortical manifold, thereby improving its generalization performance. These results suggest a dual role for cerebellar signals in motor adaptation. Firstly, these signals translate preparatory cortical activity into adapted motor output and their absence triggers cortical compensatory mechanisms that are evident already during motor preparation. Secondly, cerebellar signals constrain cortical preparatory activity into a low-dimensional manifold by introducing the task-structure information needed for generalization both at the neural and behavioral levels.

## Results

### Cerebellar block impairs force field adaptation

We trained two monkeys to wear an exoskeleton (Kinarm system) and perform an instructed delay center-out planar reaching task. The two monkeys were implanted with a chronic stimulation electrode in the superior cerebellar peduncle (SCP), which was used to block the outflow of cerebellar signals by applying high-frequency stimulation (HFS), as reported in detail elsewhere[31]. The task design (Fig. 1a, b) consisted of four different task conditions: two pre-learning conditions with and without cerebellar outflow block (termed control and HFS, respectively), and two force-field learning epochs (termed FF and FF-HFS respectively). During the control trials, the monkeys were able to maintain straight trajectories towards the presented target (Fig. 1c). In the presence of HFS, the trajectories became more variable, as quantified by their maximal deviation from a straight line (t-test: p = 0.00072 Monkey P, p = 2.1871e-10 Monkey S). Figure 1d presents trajectories of movements during a single FF adaptation session and the effect of adding HFS on adaptation. In both cases, the FF caused deviations in arm trajectories, but when the FF adaptation was combined with HFS, the trajectories were substantially more variable, even though the monkeys were able to reduce the error and converge to a partially-adapted state (similar to cerebellar patients[32]). The trajectory errors caused by the FF were quantified by measuring the vertical distance from the shortest (straight) path connecting the initiation and end points of the performed trajectory. The error was measured at two time points along the trajectory: an early time point (150 ms after movement onset), at which feedback was not yet available, and the time of maximal deviation. Figure 1d depicts the learning curves during FF (blue traces) and during FF-HFS (red traces) as measured at the early (left panel) and late (right panel) time points of the trajectories. In both conditions, learning was initially fast and replaced by a slower learning phase after about 10 trials. In terms of maximal deviation, both phases of adaptation were impaired by the HFS (2- way ANOVA, HFS effect, p = 4.04e-56; trial number effect, p = 4.18e-97, Interaction effect, p = 2.4e-4).

Finally, the significantly impaired adaptation under HFS was demonstrated using catch trials, where the FF was unexpectedly removed (Fig. 1e). Catch trials are used to expose the underlying learning achieved during adaptation trials[34]. The learning index, calculated using the catch trials (see Methods), was divided into bins of 10 trials each (Fig. 1f). In the presence of HFS, the learning index was significantly lower than in the control trials (2-way ANOVA, HFS effect: p = 0.0365, Time bin effect p = 0.0089 for the deviation measured at 150; HFS effect p = 0.0028 and Time bin effect p = 0.0018 for maximal deviation). This lower learning index further confirmed the observed impaired adaptation when cerebellar signals were blocked.

### Blocking cerebellar outflow decreased error sensitivity and increased internal noise

The efficiency of motor adaptation is driven by two opposing processes: *error sensitivity*, which measures the extent of learning from previous errors, and the *retention factor*, which quantifies the ability of the system to retain the newly acquired motor skill[2,35,36]. We calculated these two measures from our data (see Methods section) and found that under HFS, the error sensitivity decreased, on average, by 22% (paired t-test, p < 0.0001) whereas the retention factor decreased by 5% (paired t-test p < 0.005), suggesting that the impaired adaptation under HFS is mostly driven by impaired learning from previous errors (Fig. 2a, b).

Next, we explored which mechanisms were most likely related to the decrease in error sensitivity under HFS. Error sensitivity is strongly affected by execution noise through its relationship to error size and error consistency. Behavioral studies have shown that repeated exposure to inconsistent errors (i.e., errors that fluctuate in polarity around the mean trajectory[37]), or variable perturbations (i.e., irregular cursor-to-hand angular deviations on a visuomotor rotation task[38]) lead to a reduction in error sensitivity and impaired adaptation. It is, therefore, conceivable that the drop in error sensitivity under HFS was simply the outcome of the increased movement variability in these conditions. To address this possibility, we estimated the conditioning-specific motor noise by calculating the variability of the trajectory errors (i.e., the variance of the maximal deviation values) in the pre-learning conditions (i.e., during control and HFS trials). The results showed that under HFS, motor noise was significantly higher than in the control trials (Fig. 2c, paired t-test, p < 0.0001).

Nevertheless, the added execution noise under HFS could not fully account for the reduced error sensitivity. To show this, we first selected sessions with similar levels of noise during the control and HFS trials and measured the error sensitivity during FF and FF-HFS adaptation sessions. Figure 2c depicts a selected subset of sessions with matched noise levels (at a ratio of 0.6 to 1.4 between the corresponding noise levels), and Fig. 2d presents the mean error sensitivity calculated for this subset. We found that during FF-HFS, error sensitivity was significantly lower than in the FF trials (paired t-test, p < 0.0001) even though the noise levels were comparable.

In addition, the sheer magnitude of the noise level (irrespective of the similarity in noise level between control and HFS) was also unable to explain the changes in error sensitivity during HFS. This was demonstrated by classifying the recording sessions into low or high levels of noise (Fig. 2e) and then calculating the mean error sensitivity in 4 (2 × 2) different conditions: low/high noise x FF/FF-HFS (Fig. 2f). We found that error sensitivity was generally higher at low vs. high levels of noise, but within each noise level category, the error sensitivity was significantly lower in the presence of HFS (Fig. 2f, results for the two monkeys pooled, HFS effect: p = 5.47e-13, noise level effect: p = 3.72e-32, Interaction effect: p = 1.41e-4).

Finally, motor noise is often categorized as either internal (i.e., execution noise) or external (a consequence of external interferences in the environment). Since external (but not internal) noise reduces error sensitivity[38], it was crucial to identify whether the added motor noise in the HFS trials was treated as an external noise that will trigger a corrective (adaptive) response in subsequent trials or an internal source of noise that will be ignored. Using this approach, we found no adaptive response under HFS (in the absence of FF), as evidenced by the lack of correlation between the error size on a given HFS trial and

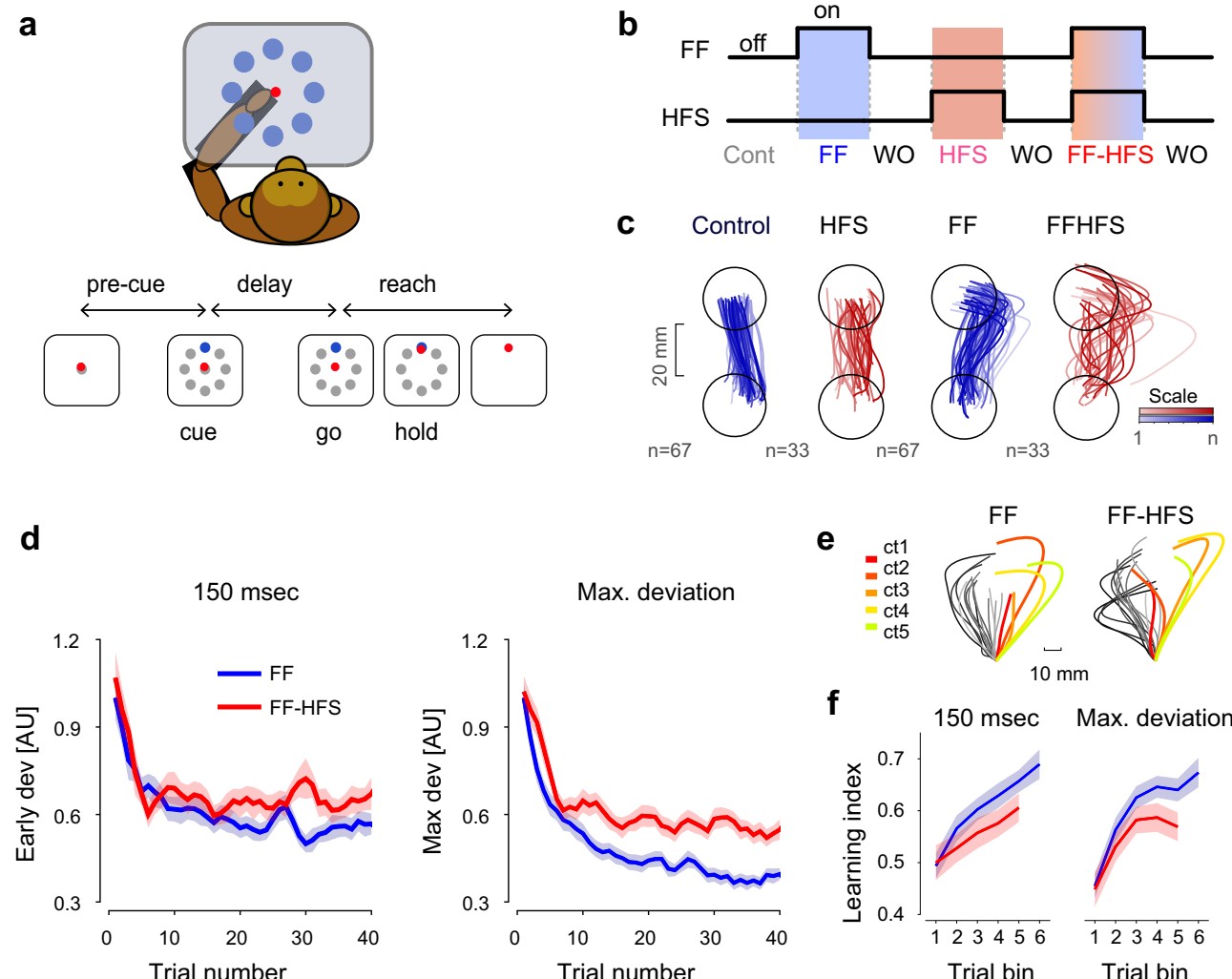

**Fig. 1 | Experimental setup and behavioral results. a** Behavioral paradigm. Monkeys were trained to wear an exoskeleton and control a cursor that appeared on a horizontally positioned screen. The sequence of events composing a single trial included a pre-cue period, onset of a peripheral target ('cue'), a delay period, and a 'go' signal (removal of the central target), after which the monkey had to acquire the target within a limited response time for a liquid food reward. **b** Sequence of conditions composing a single session, including control (Cont.) trials, force field (FF) trials, washout (WO), and HFS trials. Blue and/or red shadings indicate application of FF and/or HFS. **c** Hand trajectories in each of the four conditions (±FF and ±HFS). Color hues of each trajectory correspond to the time during the session in which the trajectory was performed (dark hues = later trials). **d** Mean deviation of the hand trajectory from the shortest (straight) line connecting the initial and the end position as a function of trial number during FF (blue) and FF-HFS (red) trials. Deviations were calculated at 150 ms after movement onset (left)

and at the point of maximal deviation (right). Data are presented as mean values ± SEM. Single session data were normalized (see Methods) and deviations are shown in normalized units. **e** Trajectory samples taken from a single session during FF (left) and FF-HFS (right) conditions. Color-coded trajectories highlight the catch trials that were interleaved between FF trials (gray trajectories - dark to light shadings correspond to early to late trials, respectively). The color of the catch trial trajectories corresponds to the time in the session in which trials were inserted on a red-to-green scale (red - early trials, green - late trials). **f** Learning index during FF (blue) and FF-HFS (red) trials plotted as a function of the trial bin (groups of 10 trials) for early (left) and maximal (right) deviations. In one monkey FF sessions were longer than FF-HFS sessions (see Methods section) and therefore the blue curves span more trial bins than the red curve. Data are presented as mean values ± SEM. Source data are provided as a source data file.

the change in error between flanking trials (Fig. 2g, regression slope in control −0.015, n.s.; regression slope in HFS −0.016, n.s. - pooled data for both monkeys, see Supplementary Fig. S1 for individual monkey data), whereas FF produced a seemingly significant adaptive response (Fig. 2h, regression slope in control −0.015 n.s.; regression slope in FF −0.097 p = 8.3e-15 - pooled data for both monkeys, see Supplementary Fig. S1 for individual monkey data). This result suggests that HFS likely acts as an internal noise source, affecting adaptation by mechanisms other than the addition of an external (i.e., perceivable) noise source.

Taken together, these results suggest that the impaired adaptation during HFS was driven by two (at least partially) independent processes: decreased error sensitivity and increased internal noise.

## The cerebellar block induced target-independent and target-specific changes in the preparatory cortical state

We recorded neural activity from the primary and premotor cortical areas (Supplemental Fig. S2) during task performance and tested the effects of FF and/or HFS on cell activity (Fig. 3a). At the population level, the cerebellar block led to a decrease in the task-related modulation of firing rates, measured by comparing the maximal rate modulation in HFS to the control (Fig. 3b, paired t-test, n = 218 p < 3.02e-6 for both monkeys, p < 2e-3 for monkey P and p < 5.4e-3 for monkey S). The effect of HFS was apparent even before the "Go" signal during the preparatory period, suggesting that cerebellar output affects motor cortical firing even before movement starts. Since

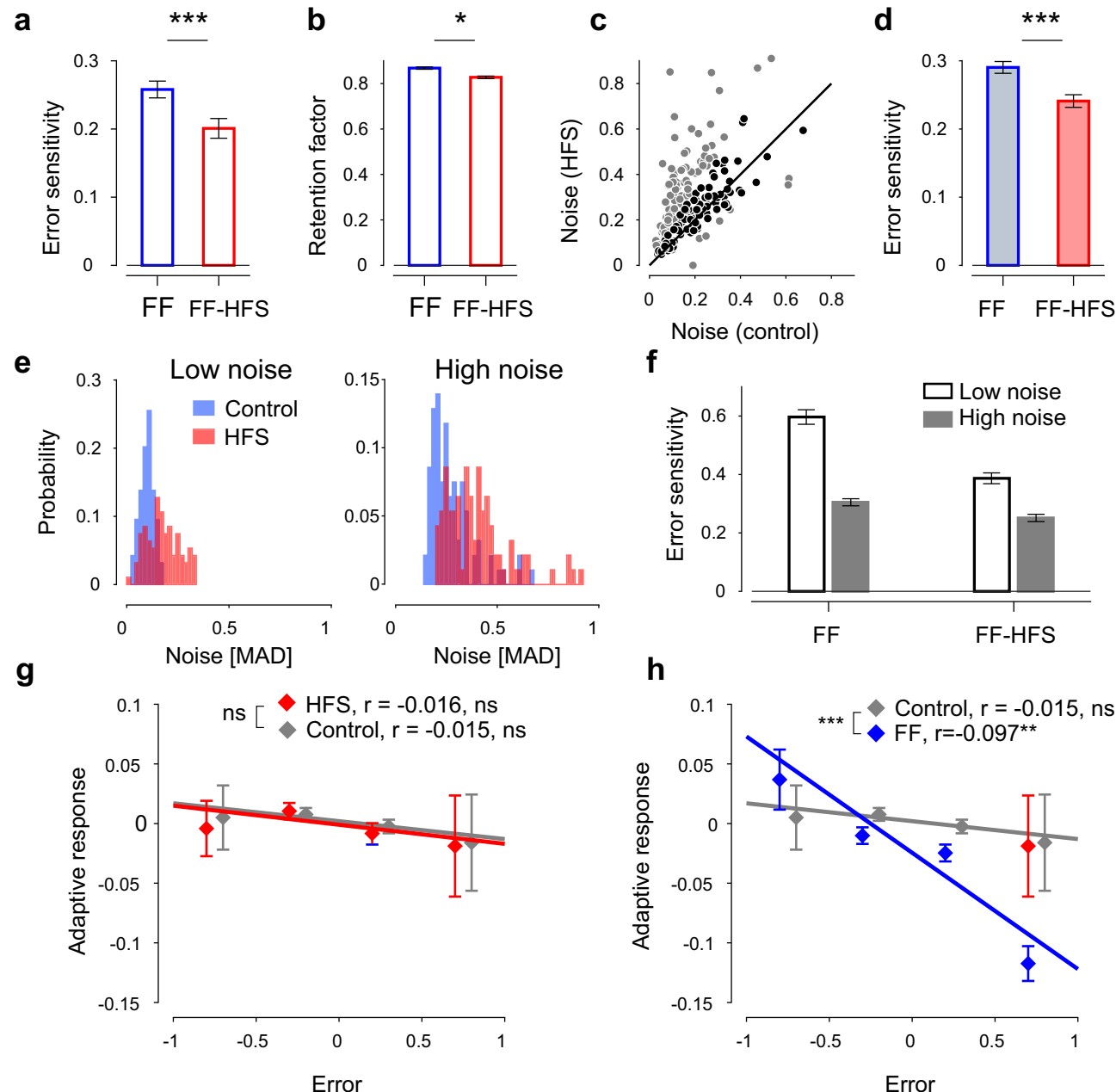

**Fig. 2 | Effect of HFS on adaptation rate. a** Error sensitivity during adaptation for FF (blue) and FF-HFS (red) conditions, averaged over all sessions (n = 191) and pooled across the two monkeys. Data are presented as mean values ± SEM. **b** Average retention factors ±SEM calculated for the same sessions as in (**a**). **c** Single-session motor noise was estimated by calculating the mean absolute deviation (MAD) of maximal deviations during HFS trials and plotting against the motor noise calculated during the matching control sessions (n = 191). Darker dots (n = 91) indicate sessions where the motor noise under HFS was comparable to the motor noise level during the control trials (i.e., HFS/Control ratio >0.6 and <1.4). **d** Mean error sensitivity ±SEM calculated for a subset of adaptation sessions (n = 91), for which the pre-adaptation noise level was comparable during FF (blue) and FF-HFS (red) conditions (i.e., highlighted sessions in **c**). **e** Distribution of motor noise in each session during the control (blue) and HFS conditions (red) for low (left) and high (right) noise levels. Sessions were split into low-noise (n = 94) and high-noise (n = 93) sessions based on the median noise level (see Methods). **f** Error sensitivity in low-noise (white bars) and high-noise (gray bars) sessions (as defined using the control and HFS trials) and calculated in the FF and FF-HFS conditions. For each condition (low-noise, n = 94 and high-noise, n = 93) data are presented as mean error sensitivity ±SEM. **g** Adaptive response, quantified by the difference in maximal deviation between trial n + 1 and n-1, regressed onto the deviation on n-th trial (i.e., Error). The data from the two monkeys were pooled (number of observations in control = 5250 and in HFS = 4906). In each bin (i.e., range of error values) the mean adaptive response ±SEM is shown. **h** Same as (**g**), for the adaptive response in the FF conditions (number of observations = 9767), and compared to control. Source data are provided as a source data file.

previous studies showed that motor adaptation is accompanied by changes in the coordinated activity of motor cortical neurons during motor preparation[25,35], we tested whether cerebellar signals are involved in recalibrating movement by shaping motor cortical preparatory activity in a task-dependent manner. To do so, we first applied

dimensionality reduction methods on the pre-learning dataset to identify the target-related information available during preparatory activity in the control conditions (Fig. 3c). As in previous studies, we found that within the subspace spanned by the first three principal components, preparatory activity was radially organized according to

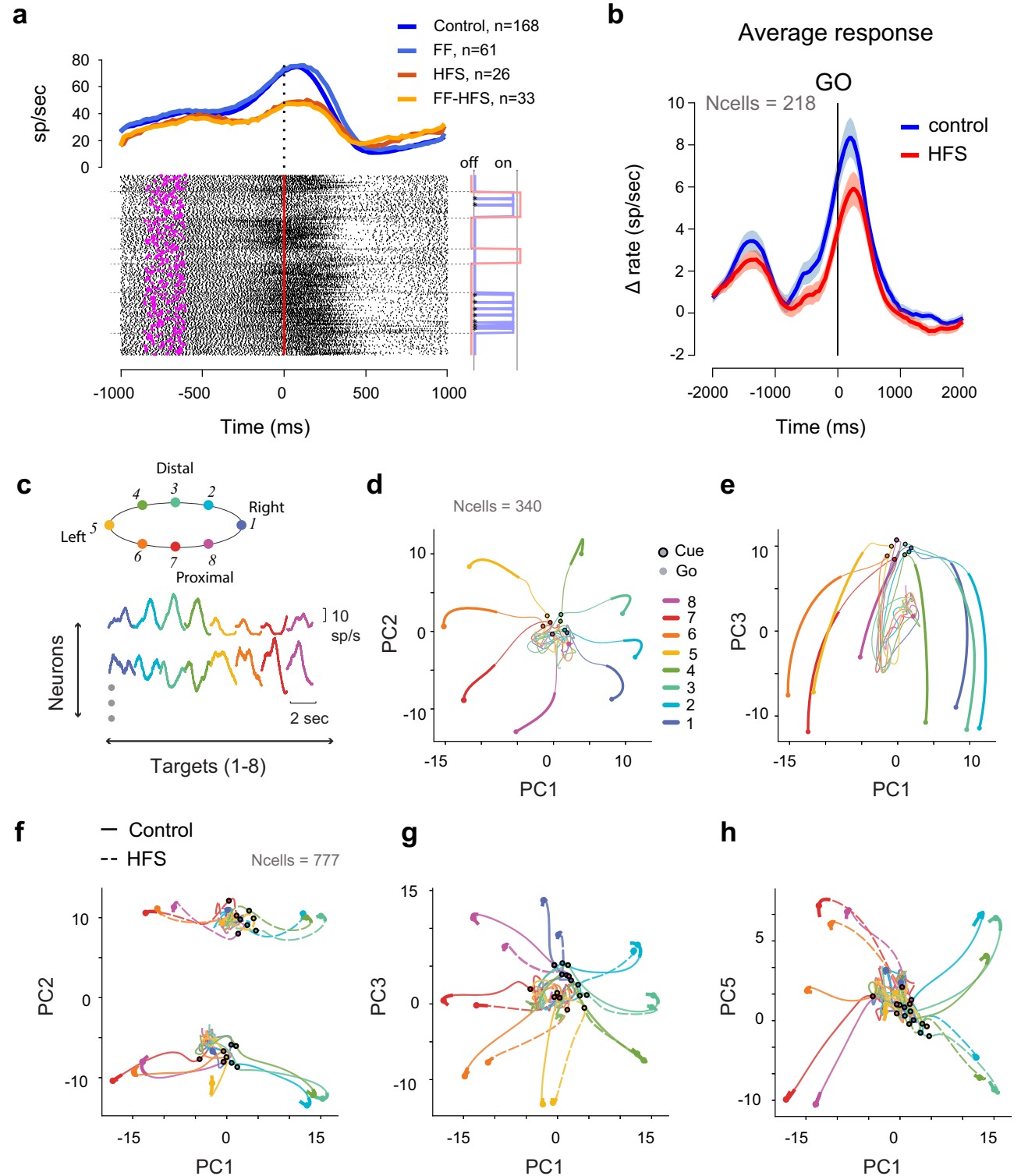

the target-related reach direction (Fig. 3d) and trial time (captured by PC3, Fig. 3e). A similar constellation was found when examining the control and HFS data when the monkeys performed an 8-target reaching task (see Methods), where we found both target-independent (Fig. 3f, g with similar results for PC4, data not shown) and target-dependent (Fig. 3h) effects of HFS on neural states compared to the control conditions. Specifically, while the target-independent effects appeared in some cases as a uniform shift in firing rates between the two conditions (e.g., Fig. 3f) the target-dependent effects distinguished between distal targets located away from the body (blue-shaded targets in Fig. 3h) and proximal targets located near the body (red-shaded target), suggesting that the cerebellar signals contained target-related information.

### Preparatory activity was more sensitive to force field perturbation during the cerebellar block

During the adaptation trials, although the monkeys only experienced the force field when movement started, their pre-movement activity was considerably affected by the subsequent events. We analyzed the neural states for the four different conditions (±HFS, ±FF) during

**Fig. 3 | HFS alters neural activity in both a target-dependent and independent manner. a** An example of task-related activity of a single motor cortical cell. All trials were directed towards the learned target but under the four conditions (±HFS x ±FF). Bottom: raster plot aligned to the 'Go' signals (time 0); magenta symbols denote cue onset. Right: onset of FF (blue) and HFS (red) conditions; Asterisks denote catch trials. Top: trial-averaged firing rates in the four conditions. **b** Average task-related activity of all cells during the control and HFS conditions (64 cells from monkey P and 154 cells from monkey S. All cells were directionally tuned and were recorded during all 4 test conditions (±HFS x ±FF). Single-cell PSTH was calculated around the 'Go' signal (50 ms bin size, and normalized by subtracting the pre-cue firing level. The shaded area depicts the standard error of the mean. **c** top: targets arrangement in the setup (indicating target position relative to the monkey). Bottom: illustration of data matrix used for the PCA analysis. For each cell, we concatenated the target-specific response (−1 s to +1 s around the 'Go' signal), forming an $n \times 8 \cdot t$ matrix, where $n$ is the number of neurons and $8 \cdot t$ is the number of time bins spanning the response to the eight targets. **d** Projection of single-cell activity from 'cue' (black circled symbols) to 'go' (filled circles), projected on the first 2 PCs, color-coded by target. Thick lines indicate the epoch used for calculating the PCA. The explained variance for each PC was: PC1:0.32; PC2: 0.17; PC3: 0.11; PC4: 0.08; PC5: 0.06. **e** Same as d but projecting on PC1 and PC3. **f, g** Effects of HFS on coordinated neural activity, calculated to a data set with reach to 8 targets for both control and HFS conditions (see Methods). PCA was performed in a similar manner as in (**d, e**), concatenating control (solid) and HFS (dashed) data (explained variance: PC1:0.2; PC2: 0.16; PC3: 0.1; PC4: 0.07; PC5: 0.07). **f** PC1 and PC2 **g** PC1 and PC3. **h** PC1 and PC5. PC4 did now show HFS dependent difference (see text). Source data are provided as a source data file.

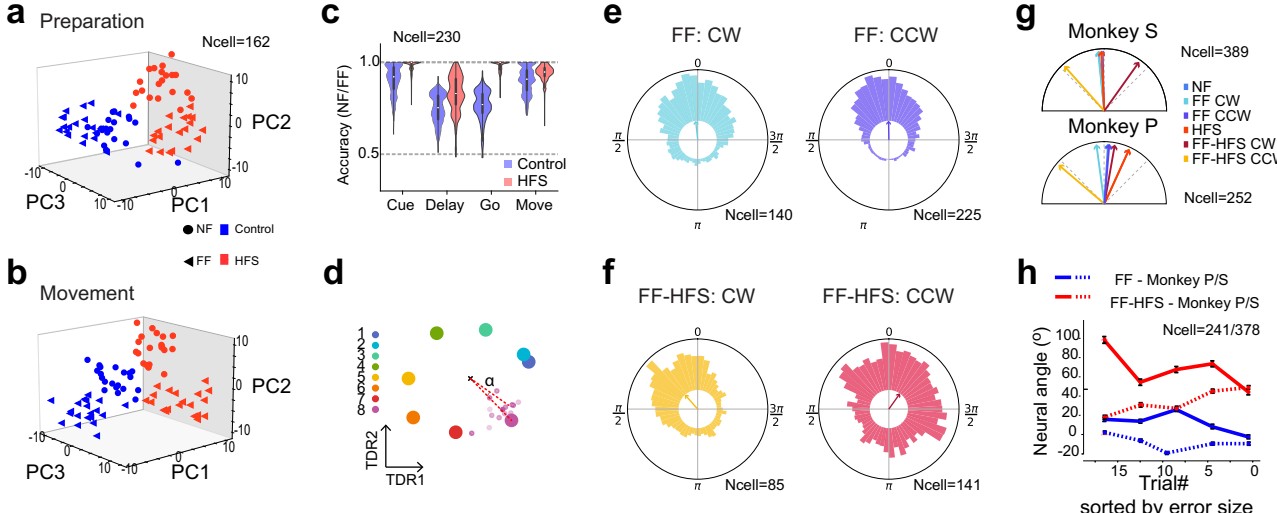

**Fig. 4 | Preparatory neural activity during HFS is more sensitive to the subsequent force field. a** Projection of trial-specific coordinated activity on the first 3 PCs during movement preparation. Each dot presents coordinated activity in a single trial during the sequence of trials performed in the control (blue) or HFS (red) conditions, either before adaptation (null field, NF - circles) or during adaptation (FF - circles). Neural data were averaged across all targets, during a time window spanning −300 ms to 0 around the Go signal. **b** Same as (**a**) but calculated during the 100–400 ms after the Go signal. **c** Decoding accuracy of adaptation conditions (NF vs. FF) based on epoch-specific data for the control (blue bars) or HFS (red bars). Dashed line denotes chance level (0.5). Decoding accuracy values were obtained for different training and testing sets. **d** Projection of target-specific neural activity 100 ms before the Go signal on the TDR axes. Large dots correspond to the trial-averaged activity used to compute the TDR axes, for each of the 8 targets. Small dots represent the activity in different trials associated with the 8th target in the Control condition. The neural angle denotes the angle of a trial's projection relative to the projection of the mean activity in the Control conditions for the same cued target. **e** Polar histograms of the neural angles calculated for one monkey (monkey S) by aggregating data from all cued targets and trials in the control FF conditions with the same force field direction (left: clockwise CW trials, right: counter-clockwise CCW trials). **f** Same as (**e**) but during FF combined with HFS (FF-HFS). **g** Mean neural angles calculated for different force-field and HFS conditions, for monkey S (top) and P (bottom). **h** Average neural angle for each trial sorted by trajectory errors (ordered from high to low maximal deviation). Neural angles for monkey P/S are shown respectively in full and dashed lines for FF (blue) and FF-HFS (red) conditions. Figure includes data for targets 4 and 8 which required inter-joint coordination. Data are presented as mean values ± SEM. Source data are provided as a source data file.

movement preparation and execution (Fig. 4a, b and Supplementary Fig. S3). Since in any given learning session, all trials were directed towards a single target, we averaged the neural activity across targets to obtain the target-independent neural states. The effects of HFS and FF were clearly evident during movement when all four conditions were equally separable (Fig. 4b). Notably, however, the four conditions were already separable during movement preparation (Fig. 4a) but to varying extents, such that the separability between subsequent FF and null field (NF) trials was larger under HFS (red symbols) than in the control conditions (blue symbols). This property was quantified and verified by the decoding (FF vs. NF) accuracy (Fig. 4c), which showed that even though the noise level was the same in the two conditions, FF vs. NF decoding during the delay period was more accurate under HFS. Taken together, these results suggest that preparatory neural states encode the upcoming conditions of the trial (FF vs. NF), and that this sensitivity is accentuated during cerebellar block, although motor adaptation under these conditions was impaired.

## Changes in cortical activity within the target-related plane may drive residual adaptation under HFS

The augmented sensitivity of preparatory states to FF under HFS could underscore the adaptation-related mechanisms employed in these conditions. We tested this possibility by measuring changes in neural activity in the target-related subspace correlated with the trial-to-trial adaptation to FF. We first applied targeted-dimensionality reduction (TRD)[25,39] to the pre-learning trials, which identified the plane in neural activity that best predicts the position of the targets (Fig. 4d). We then projected the condition-specific (FF and FF-HFS) single-trial activity onto the plane spanned by the first two dimensions of the TDR and measured its angular deviation from the learned target. We found that

during FF adaptation, the angular deviations of single trials were centered around the learned targets, regardless of the FF direction (clockwise or counterclockwise, Fig. 4e). Conversely, under FF-HFS, the single trial angles had a substantially higher angular variance, with a significant rotation of the mean bias consistent with the FF direction (Fig. 4f, g). Testing for a correlation between the mean angular deviation and the adaptation process (Fig. 4h and Supplementary Fig. S4) revealed how the gradual change in the angular deviation as the error decreased. Specifically, during FF adaptation, when trajectory errors were large (corresponding to early FF trials), neural activity only evidenced a small deviation that continued to diminish throughout the adaptation process such that when the errors were small (corresponding to late adaptation trials), the neural state was nearly aligned with the learned target (Fig. 4h, blue lines). In stark contrast, under FF-HFS, the neural angle settled when adaptation was completed at around 45 degrees (Fig. 4h, red lines). For monkey P (which moved slower than monkey S) we only used targets that required inter-joint coordination, but similar results were obtained when utilizing the same targets in monkey S (Supplementary Fig. S4). This result suggests that the large separation in trajectories between NF and FF conditions under HFS may at least partially reflect an adaptive rotation of the neural angle away from the learned target. The cortical change in neural angle may reflect a "re-aiming" policy used by the motor system as a compensatory mechanism aimed to handle the force field in the absence of online cerebellar control.

### Cerebellar signals induced task-relevant low-dimensional cortical representations

If the rotational shift in cortical preparatory states during FF-HFS represents a cortical compensatory mechanism, it could mean that cerebellar signals are not involved in shaping motor preparatory activity, at least in NHPs performing an upper-limb reaching task, but function only as an online controller of movement. In this case, the preparatory state during null field conditions (when no cortical compensation is required since there is no adaptation) should not be affected by the HFS. To address this question and to quantify the effect of cerebellar signals on motor preparation, we calculated the effective dimensionality of cortical activity in the control and HFS conditions. To measure dimensionality, we calculated the participation ratio[40,41] of the condition-average neural activity (see Methods) obtained when the monkeys performed 8-direction reaching movements in the control and HFS conditions. The a priori assumption is that the participation ratio should range from 2, indicating that the trial-averaged activity of all the targets lie exactly on a single plane, to 7, indicating that the neural representations of the eight targets are uncorrelated, thus defining a 7-dimensional subspace. We found a significant increase in the dimensionality of preparatory activity under HFS as compared to control conditions (Fig. 5a and Supplementary Fig. S5). In contrast, neural activity before the Cue and after the Go events had a similar dimensionality (high and low, respectively) in the control and HFS. This result is consistent with our previous report of decorrelated activity during HFS[31]. Importantly, the increased dimensionality under HFS was not driven by noise. The trial-to-trial variability measured by the Fano factor did not differ between the HFS and the Control conditions (Supplementary Fig. S6).

To show that cerebellar output can indeed modify the dimensionality of motor cortical signals and the functional consequences of this effect, we implemented a simple computational model of cerebellar-to-cortical interactions (Fig. 5b). In this model, the cortical network integrated two kinds of inputs: a high-dimensional target-related command and low-dimensional cerebellar feedback. The low-dimensional feedback is "task informed", since it conveys information about the spatial arrangements of the learned targets relative to each other, which in our case corresponds to a ring arrangement on a plane. Although there was no learning in the modeled motor cortex and the

cortical synapses were considered random, the low-dimensional feedback was sufficient to quench motor cortical activity into an effective lower-dimensional neural manifold, as measured by the participation ratio (i.e., Dpca). Further, a reduction of the feedback gain resulted in an increase in dimensionality, consistent with the increased dimensionality we found experimentally when blocking cerebellar output (Fig. 5c).

### The increased dimensionality of cortical activity during HFS was accompanied by a decrease in generalization performances

The increased effective dimensionality during the preparation period suggests a loss of structure in the neural representation and lower correlations between target-specific neural representations. Previous studies have noted that low dimensional task representations supported task abstraction, which is needed for generalization[42,43]. The ability to utilize an abstract representation for generalization can be measured by the Cross-Condition Generalization Performance (CCGP) of a linear readout[42]. To demonstrate how dimensionality and structure enable generalization, one can consider a dataset generated from a ring embedded in a high-dimensional space. After dividing the ring into four quadrants, it is possible to train a linear classifier to separate data taken from top and bottom halves of the circle on the left side and test its performances on data taken from the right side of the ring. If the ring is perfectly embedded within a two-dimensional subspace, the effective dimensionality is D = 2, and the classification would generalize well. On the other hand, if the ring is embedded through a nonlinear transformation and does not lie within a single plane, the effective dimension would increase, and the generalization performance would diminish[42] (Fig. 5d). Thus, the CCGP can measure the generalization performance across different possible dichotomies of the ring. Following this argument, we quantified the task abstraction in the cortical representations by calculating the CCGP during the preparatory period. As expected, the increased dimensionality in the presence of HFS was accompanied by a decrease in the CCGP (Fig. 5e). This result was also replicated in our simplified model where the decrease in the low-dimensional feedback resulted in increased dimensionality and decreased CCPG (Fig. 5f).

Finally, we tested whether the increased dimensionality and reduced generalizability of task-related neural activity under HFS would also extend to changes in behavior. To this end, we tested the ability to generalize learned behavior to untested targets under HFS. Specifically, a monkey was required to implement an adapted response acquired from the learned target to movements directed to neighboring targets (Fig. 6a, b). We found that blocking cerebellar output using HFS impaired the ability to generalize target-specific adapted behavior. The effect of HFS was specifically pronounced when generalization trials presented late on the adaptation block when adaptation to the learned target was completed (2-way Repeated Measures ANOVA, stimulation × time interaction, p = 7.4e-3). Late, but not early, generalization was significantly impaired compared to the control condition (t-test p = 0.022) (Fig. 6c–e). These results are aligned with previous models, suggesting that the cerebellum encodes task-specific information[44]. Taken together, these results suggest that cerebellar signals may play a dual role in regulating FF adaptation and generalization through their specific roles in motor execution and motor preparation.

### Discussion

The cerebellum plays a key role in motor adaptation[1,2,45], but neural correlates of adaptive responses can also be found in the motor cortex as early as the preparatory period[22,25,35]. These early cortical signals may be an outcome of cerebellar-to-cortical interactions[28–30] or evolve locally through intracortical processes. Here we addressed this question by reversibly blocking cerebellar outflow while monkeys performed a force-field adaptation task. We found that this manipulation

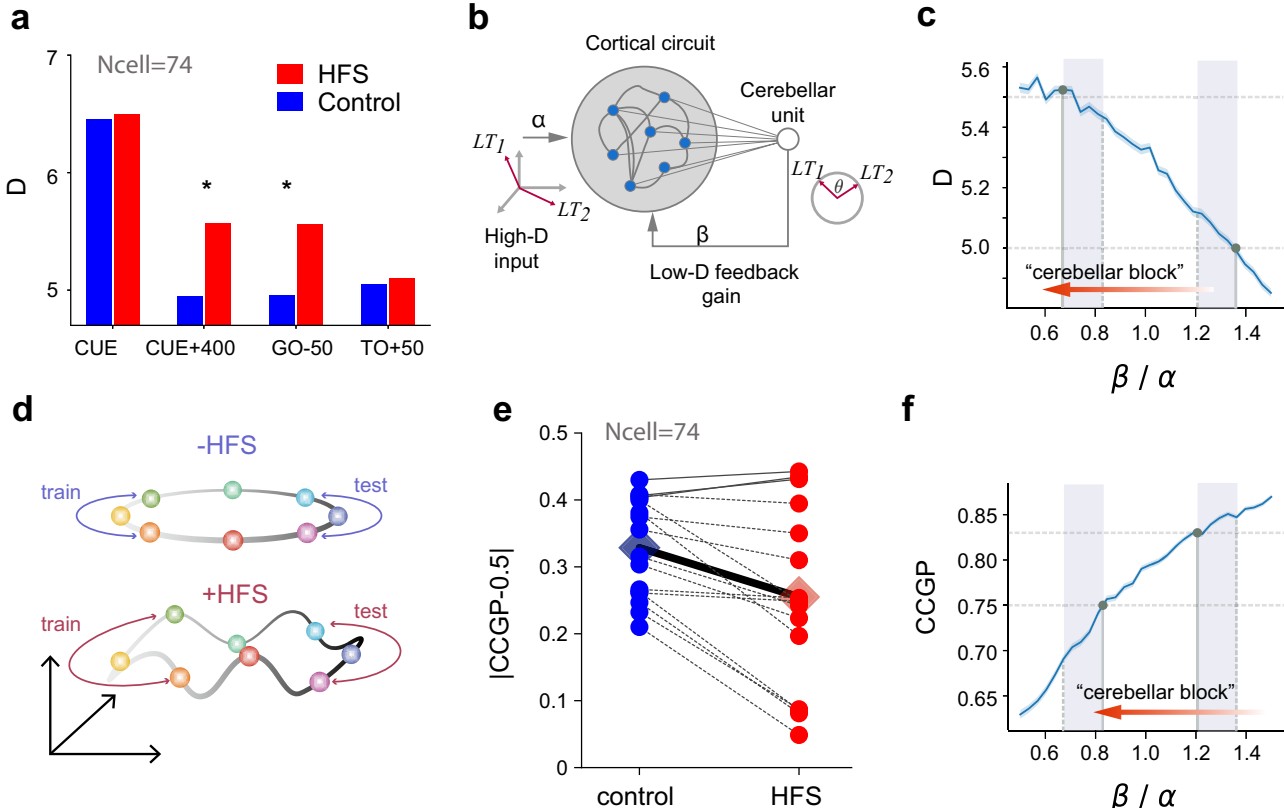

**Fig. 5 | Cerebellar signals constrain the dimensionality of preparatory activity and improve generalization. a** Dimensionality of neural activity estimated by the participation ratio (see Methods) at different epochs (blue bars: control, red: HFS). Asterisks denote significant differences in dimensionality ($D_{pca}$) during control vs. HFS conditions (resampling test with n = 1000, p < 0.01) **b** A simple network model captures the effects of the cerebellar feedback on motor cortical activity. The recurrent network integrates the external high-dimensional command signal (with gain $\alpha$) and the low-dimensional feedback (with gain $\beta$). At input, targets are uncorrelated and span an 8-dimensional subspace. At feedback, targets are embedded in a low-dimensional ring and are represented by a single variable $\theta$. **c** $D_{pca}$ (participation ratio) calculated for the modeled cortical activity (as in **b**) as a function of the low-dimensional feedback gain relative to the high-dimensional input ($\beta/\alpha$). LT1 and LT2 are examples of learned targets represented in the high-dimensional input and the low-dimensional cerebellar embedding. Horizontal dashed lines show the $D_{pca}$ obtained for the monkey data in control (lower level)

and HFS (higher level) conditions. Solid vertical lines show the $\beta/\alpha$ value corresponding to the intersection point with the $D_{pca}$ (circles). Vertical dashed lines show the matching $\beta/\alpha$ values obtained when calculating CCGP (panel f below). The shaded area shows the low and high $\beta/\alpha$-ranges that best fit the data with and without cerebellar block. The red arrow shows the gradient of $\beta/\alpha$ simulating the cerebellar block. Simulation parameters: external gain $\alpha$ = 0.25 and intracortical gain g = 1.75. The shaded area around the curve denotes the standard error over 600 network realizations. **d** Illustration depicting the effects of the topology of the neural representation on cross-condition generalization performance (CCGP). **e** CCGP relative to chance level (0.5) calculated for Control (blue) or HFS (red) conditions. Each dot represents a pair of dichotomies with shared symmetries (e.g., top-down dichotomy on left targets and top-down dichotomy on right targets). Diamonds represent the average CCGP over all dichotomies. **f** Similar to c but for the CCGP computed for the network model as a function of the low-dimensional feedback gain. Source data are provided as a source data file.

led to increased motor noise and impaired adaptation that were accompanied by two notable changes in the preparatory neural state. First, in FF-HFS, neural trajectories calculated before movement onset shifted away from the learned target, whereas during FF trials, the neural trajectories remained aligned. Second, under HFS, the dimensionality of preparatory activity increased, leading to a more complex and less generalizable neural space. This outcome was explained by a computational model that showed that a low-dimensional cerebellar-like input can lead to a cortical network with reduced dimensionality. In addition, the model predicted that the loss of cerebellar signals would lead to impaired generalization, as confirmed experimentally in the reduced generalization under HFS at both the neural and behavioral levels.

These adaptation-related motor impairments under HFS closely resemble the deficits found in cerebellar patients. The adaptation to the force field under HFS was impaired (Fig. 1e, g) but not completely abolished. Similarly, adaptation in cerebellar patients to a repeatedly presented target was shown to be maintained even in severely affected subjects[32,46–48]. In addition, the poor adaptation during HFS was driven

by low error sensitivity (Fig. 2a), reflecting the lesser capacity of the system to learn from previous errors[38], and increased motor noise as measured by the trial-to-trial variability of hand trajectories (Fig. 1c). Augmented motor variability is a core symptom in cerebellar patients[15,46,49–52], although it was shown that motor noise alone is insufficient to explain the reduced error sensitivity and the subsequent impaired adaptation[33]. Likewise, the low error sensitivity under HFS was maintained even when matching the error size to the control conditions (Fig. 2d), thus indicating that without cerebellar signals, the ability to learn from previous errors is fundamentally disrupted. Taken together, these results show that HFS accurately replicated both the difficulties exhibited by cerebellar patients when trying to adapt to an external force field[49,53,54] and the underlying mechanisms leading to these impairments. However, since the HFS probably blocks information flow to other cortical areas targeted by cerebellar signals (e.g., the posterior parietal cortex[55,56]), we cannot exclude the possibility that some of the behavioral impairments observed under HFS, such as increased motor noise, were mediated by these additional pathways which were not explored here.

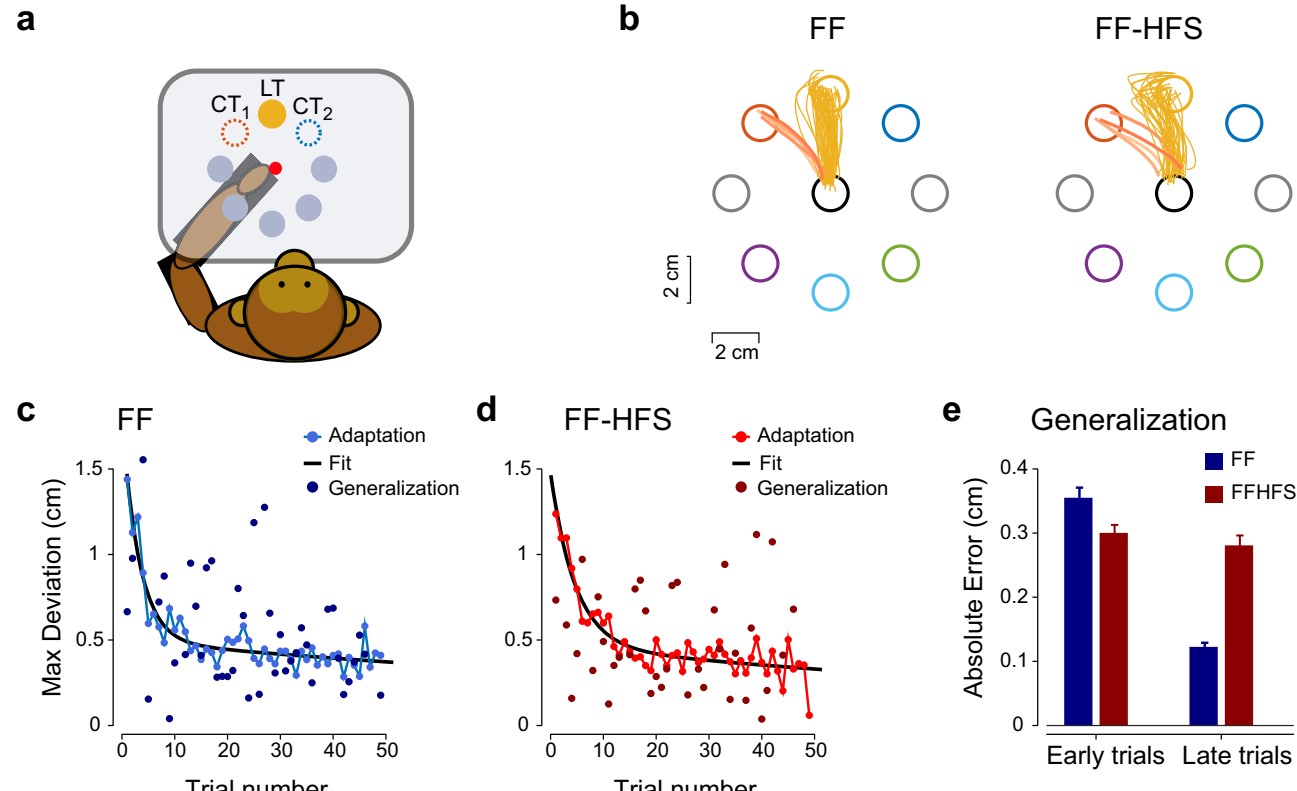

**Fig. 6 | Impaired generalization behavior during HFS trials. a** One monkey performed the same task as shown in Fig. 1a. However, during the adaptation sequences, we used randomly selected probe trials in which the monkey was instructed to move to one of two targets, flanking the learned target. This approach was used to estimate the generalization of learned behavior to nearby targets. **b** Example of one session in which probe trials were presented to the left of the learned target during FF (left panel) and FF-HFS (right panel) conditions. The hue of the probe trajectories indicates the order of trial presentation in the sequence (light/dark - early to late trials, respectively). **c** Summary of adaptation and generalization across all sessions (n = 45). The adaptation sequence (light blue lines and symbols) and a fitted adaptation curve (dark blue) are shown. The generalization samples (n = 42) show the (non-averaged) deviation in all probe trials presented at different positions during the adaptation session. **d** Same as panel i but for the FF-HFS conditions (45 sessions and 37 generalization samples). **e** Quantification of generalization across all sessions, calculated for early (1–5) and late (>= 10) trials in the FF (blue) and HFS (red) conditions (n samples 4 and 35 for early and late FF trials respectively, and 4 and 30 samples for early and late FF-HFS trials. Data are presented as mean values ± SEM. Source data are provided as a source data file.

Blocking cerebellar signals during motor adaptation trials led to a significant alteration in pre-movement motor cortical activity. The preparatory neural state under HFS was more sensitive to the subsequent force field compared to the control conditions (Fig. 4a–c), which may hint at more adequate preparation for the upcoming perturbation despite the impaired adaptive behavior in this condition. In addition, during FF-HFS trials (but not in FF trials) the neural angle of the preparatory state (calculated in the target-related subspace) rotated in a way that was correlated with the direction of the subsequent force field (Fig. 4g). By the end of the adaptation, the neural angle hovered around 45° relative to the learned target in what could be interpreted as a re-aiming strategy (Fig. 4h). Studies on cerebellar patients have also revealed asymmetries in adaptation between CW and CCW. It is argued that these patients manage to compensate for the loss of cerebellar predictive signals by using a volitional (cortically-based) strategy to leverage the directional force field to complete the task[32]. Our results are fully consistent with this hypothesis. Hence, in the absence of cerebellar signals, the motor cortex appears to be able to identify patterns in the external environment and adjust its motor plan to suit (or leverage) these patterns in an effort to reduce error. This result departs from other reports[25], which found a re-aiming strategy during FF adaptation. This outcome could be related to the differences in task conditions since Sun et al. used a haptic device and isometric wrist movements, which resulted in slower adaptation. Importantly, our findings show that adaptation to FF does not require a

rotational shift in cortical activity as long as the cortico-cerebellar circuit is intact. Previous studies of FF adaptaion[27] have shown that during movement planning, neurons in SMA but not in M1 exhibit an adaptation-related shift in the preferred direction. It is possible that the motor cortical compensation which we found in the absence of cerebellar signals is at least partially driven by SMA signals.

The involvement of cerebellar signals in shaping preparatory activity may extend beyond motor adaptation tasks. Previous studies have reported that the cerebellum is involved in the planning stage of motor control[28,29,57]. The current findings support and add to these data by showing that cerebellar signals shape cortical preparatory activity by providing task-relevant information. Our analysis of the preparatory neural trajectories in the null field conditions revealed that part of the effect of HFS was target-dependent, and involved distinguishing between distal and proximal targets (Fig. 3h), suggesting that this information is embedded in the cerebellar output normally converging onto the motor cortical network. In addition, we showed that cerebellar signals participate in regulating the dimensionality of preparatory activity, such that in their absence, this dimensionality increases significantly (Fig. 5a). This outcome suggests that the cerebellum is responsible, at least in part, for the embedding of motor cortical activity within a low-dimensional manifold[58–60]. The cerebellum can achieve this by learning the structure of the task and projecting it back to the cortex via the thalamus[44]. Consistent with this hypothesis, our computational model showed that recurrent feedback

that learned the structure of a task was sufficient to reduce the cortical dimensionality, whereas decreasing the gain of this feedback led to an increase in dimensionality as it occurred under HFS (Fig. 5c).

Our neural analysis and behavioral results further suggest that the dimensionality reduction of cortical activity to a low-dimensional manifold can affect motor adaptation. Under HFS, the increased dimensionality of the target-related manifold implies a topological change in the neural representation that disrupts the geometric relations (angles) between the targets in the neural space. In particular, the topological change impedes task abstraction, as measured by the decrease in the Cross-Condition Generalization Performance (CCGP) under HFS (Fig. 5e). The CCGP measures how well a linear readout trained on part of the manifold can generalize to unexplored regions of the manifold[42]. Our computational model confirmed this result by showing that decreasing the gain of the low-dimensional feedback to the cortical circuit not only increased the dimensionality, but also resulted in a lower CCGP (Fig. 5f). Although CCGP directly measures generalization performance in a linear classification, it may also account for impairments in generalizing adapted responses. This is because generalization of both classification and adaptation relies on symmetries in the neural representations that are broken when the dimensionality increases. In accordance with our neural observations, our behavioral analysis showed that during the cerebellar block, monkeys failed to generalize to neighboring targets, similar to previous studies showing impaired generalization in cerebellar patients[32,61]. One limitation of this approach is that our analysis of neural states cannot directly account for the observation that in a motor adaptation task, the ability to generalize decreases as the angle from the learned target increases[8,25,62,63]. It is possible that since the dimensionality of the preparatory state was already high in the control conditions ($D_{PCA} > 5$) the targets did not reside perfectly in a single plane and that this baseline topology of population activity impeded generalization to distant targets even when no HFS was applied.

Finally, our results propose a way to link between motor adaptation and generalization. It is possible that under HFS, when the dimensionality of activity increases and the task structure is impaired, adaptation can only be local, since the specific adjustments required for adapting to target-specific force fields are uncorrelated across targets. In this model, the preparatory rotation of the cortical state constitutes the implementation of this local learning. The rich set of sensorimotor inputs integrated by the cerebellum from via mossy fibers[64], as well as task-related representations such as error and reward signals from climbing fiber inputs[10,65] place the cerebellum in an ideal position for structuring the cortical representation and facilitating the embedding of the complex external world into an egocentric cortical representation, in a process akin to coordinate transformation[66]. A future combination of experimental and computational studies is needed to validate this hypothesis.

## Methods

### Behavioral paradigms
Data were obtained from 5 *Macaca Fascicularis* monkeys (females, 4.5–5.5 kg, 6–8 years old). Monkey care and surgical procedures were in accordance with the Hebrew University Guidelines for the Use and Care of Laboratory Animals in Research, supervised by the Institutional Committee for Animal Care and Use.

**Force field adaptation task.** Two monkeys were trained to sit in a primate chair, wear an exoskeleton (KINARM, BKIN Technologies) and perform a planar, shoulder-elbow reaching task (Fig. 1a). In this task, the monkeys were instructed to locate a cursor within a central target. After 500 (monkey P) or 800 (monkey S) msec (pre cue period), a peripheral target (one of 8 evenly distributed targets) appeared and the monkey had to wait another 600–850 (monkey P) or 450–700 (monkey S) msec (delay period) until the central target disappeared

("go" signal) and reach the cued peripheral targets within 800 (monkey P) or 900 (monkey S) msec. If the monkey moved the cursor to the correct target within this time frame it was rewarded with a drop of applesauce.

**Eight-target task.** Monkey P and two additional monkeys (monkeys C and M) were trained to perform an 8- target reaching task in the control and HFS conditions. This task is described in detail in refs. 31,67,68. In short, the monkeys were instructed to locate a cursor within a central target. After 500 ms, a peripheral target appeared and the monkey had to wait until the central target disappeared ("GO" signal) and then reach the cued peripheral targets. If the monkey moved the cursor to the correct target within the predefined time limit, it was rewarded with a drop of applesauce. To encourage the monkey to predict the timing of the "go" signal, we limited the total time it had to reach the peripheral target to 500 ms and inserted a 200 ms grace period before the GO signal. Onset of movement within this time frame did not abort the trial.

### Surgical procedure and recording techniques
After training was completed, in a surgical procedure under general anesthesia, a recording chamber (21 × 21 mm) was attached to the monkeys' skull above the hand-related area of the motor cortex and a small chamber was positioned above the estimated insertion point of a stimulating electrode at the Superior Cerebellar Peduncle (SCP). After recovery and a re-training period, we recorded motor cortical activity extracellularly. During recording sessions, glass coated tungsten electrodes (impedance 300–800 kU at 1000 Hz) were lowered through the chamber to different cortical sites, mostly in the primary motor cortex (M1). The signal obtained from each electrode was amplified (x10K), and bandpass-filtered online (300–6000 Hz). The signal was then digitized (32 kHz) and saved to disk. Supplementary Fig. S2 presents the recording maps of monkeys S and P.

### Insertion of stimulating electrode into the SCP
A post-surgery MRI scan was used to plan the insertion trajectory of the stimulating electrode into the ipsilateral SCP through the small chamber. Then, we inserted a chronic bi-polar stimulation electrode into the SCP (NSEX100, David Kopf Instruments, impedance range of 30-60 KΩ). To verify the correct placement of the implanted electrode, we applied bipolar stimulation pulses through the electrode and monitored the evoked intra-cortical responses measured by recording electrodes inserted in the motor cortex[69,70].

### Experimental protocols
**Force-field adaptation protocol.** Each recording session was composed of two or three sub-sessions, with each session spanning 340 trials that followed a similar protocol. Each sub-session included a set of control trials directed towards all 8 target directions. Afterwards, one of the 8 possible targets was selected as the Learned Target (LT) and all the trials thereafter were directed towards this target. These trials included a set of 30–50 control trials (with no force-field and/or HFS), a set of 30–40 trials with HFS, 50–70 force field trials (FF) and 50 force field trials with HFS (FF-HFS). Each condition was separated by wash-out trials which were also directed towards the learned target but with no FF or HFS applied. These trial sets were inserted between FF and HFS sets and between HFS and FF-HFS sets. During the adaptation sequence we inserted intermittent catch trials in which the force field was unexpectedly removed. Catch trials appeared at random in about 10% of the trials. The order of epochs in each session (i.e., control, HFS, FF and FF-HFS) was fixed throughout the recording sessions and could potentially have been memorized by the animals. However, we found that learning sessions performed early in each recording session were

**Table 1 | Median number of trials per condition per animal**

| Monkey | Control 8 | Control1 | HFS | FF | FFHFS | WO |
|--------|-----------|----------|-----|-----|-------|-----|
| P | 64 | 20 | 40 | 50 | 50 | 20ᵃ |
| S | 64 | 20 | 40 | 70 | 50 | 30 |

ᵃFor monkey P, early sessions included an 8-target washout condition which spanned 64–80 trials.

different from late learning sessions (data not shown) indicating that the animals did not memorize the sequence of events or that memorization provided no significant benefits in terms of task performance. The number of trials in each condition and for each animal are shown in Table 1. A summary of movement trajectories and behavioral parameters appears in Supplementary Fig. S7 and Supplementary Table 1.

**Generalization protocol.** In one monkey (monkey P) we tested its ability to generalize the force field learned for the LT to movements directed to neighboring targets. In these experiments we used only the most distal and most proximal targets in the anterior-posterior axis (target 3 and 7 in Fig. 1a) as the learned targets. In each sub-session we selected one learned target and one of its neighboring targets (either to its left or right) as the probe target. During the adaptation set, we randomly inserted probe trials (on average, every 10 trials) in which the probe target was presented instead of the LT. The direction of the applied FF was randomly selected to be either CW or CCW.

**Electrophysiological recordings**
Glass coated tungsten electrodes (impedance 300–800 kΩ at 1000 Hz) were inserted through the chamber to different cortical sites, mostly in M1. The signal obtained from each electrode was amplified (x10⁴), bandpass-filtered (300–6000 Hz), digitized (32 kHz), and saved to disk. Recordings were made with up to 4 individually moveable electrodes (Flex-MT by Alpha Omega, Nazareth, Israel).

We used the raw neural signal and first removed the stimulation artifacts by subtracting their average profiles[69,70]. Then, we offline-sorted the cleaned signal to extract spikes of single cells (Offline Sorter™, PLEXON). Cell activity was further inspected to confirm stable trial-to-trial activity during recordings. In this study we included both well-isolated single cells and multiunit activity. Since our single cell analyses focused on the coordinated activity of neurons and required a large number of neurons, we pooled PM and M1 neurons.

Recording sites were identified based on the distance from the central sulcus and the threshold level of stimulation required for producing a noticeable muscle twitch in the contralateral arm (50 ms burst of biphasic stimulation pulses applied at 300 Hz and at an intensity <70 μA). We found that out of 640 recorded cells, 128 (20%) were located in high-threshold sites situated >5 mm anterior to the central sulcus. These premotor areas were mostly confined to the dorsal premotor cortex. The approximate recording depth was estimated based on the actual electrode location relative to the depth at which first cortical cells were encountered. The vast majority of the cells (605 neurons, 94.5%) were recorded at a depth ≤3 mm, which means that only a few neurons were recorded from deep in the sulcus.

**SCP stimulation protocol**
To block the outflow of cerebellar signals, we applied high-frequency stimulation (HFS) through a chronic SCP stimulating electrode [Nashef et al. 2019]. HFS consisted of a long train of biphasic (200 μs each phase) stimulation pulses applied at 130 Hz, and delivered at fixed intensity (ranging from 75 to 200 μA). The HFS trains were applied during a sequence of 30–50 trials either in combination with the force field (FF-HFS trials) or in the null-field conditions (HFS trials).

**Analyses of motor behavior**
**Behavioral analysis of the force-field adaptation process.** During task performance the KINARM system (BKIN Technologies, Canada) provided continuous measures of the angular velocities of the shoulder and elbow joints and the endpoint position of the working arm. We used these measures to compute the maximal velocity, movement time, curvature index, and reaction time for each trial.

During the force field adaptation trials, the motors of the exoskeleton generated a velocity-dependent curl field orthogonal to the direction of the hand velocity in a clockwise or counter-clockwise direction, according to:

$$\begin{bmatrix} F_X \\ F_Y \end{bmatrix} = k \begin{bmatrix} \cos(\theta) & -\sin(\theta) \\ \sin(\theta) & \cos(\theta) \end{bmatrix} \begin{bmatrix} \dot{X} \\ \dot{Y} \end{bmatrix} \tag{1}$$

where $F_x$ and $F_y$ are the forces generated by the robot in x and y axes, respectively, $k = 3$ Ns/mm, $\theta = \pm \frac{\pi}{2}$, and $[\dot{X}, \dot{Y}]$, is the hand velocity vector in the horizontal plane. When applied, the curl force field changed the arm dynamics and caused a deflection of the movements from the straight trajectory line.

**Measuring the adaptation process.** For each trial $n$ we calculated the motor response $r^{(n)}$, which we defined as the maximal deviation of the trajectory from a straight line connecting the initiation and end point. We also measured the deviation at an early time point (150 ms after movement onset). To compare the adaptation curves between control and HFS trials, for which movement velocity varied, single session data were normalized by dividing the trial-specific trajectory errors by the error measured for the first trial. This forced all single session adaptation curves to start at 1. To analyze the differences between adaptation during FF and FF-HFS, we divided the deviation data into four bins of 10 trials each and applied two-way ANOVAs with HFS (+/-) as one factor and adaptation bin as the second factor.

**Calculating the learning index.** To quantify the accumulating effect of adaptation, we calculated the *Learning Index* (LI)[34]. First, we calculated the net response by discounting the natural tendency of the monkey to produce stereotyped movement $\delta r^{(n)} = r^{(n)} - \langle r \rangle$, where $\langle r \rangle$ denotes averaging the response over 20 control trials to the learned target, performed immediately before the adaptation set. The *Learning Index* at catch trial $n$ is the absolute net response relative to the response in flanking trials

$$Learning\ Index(n) = \frac{\left| \delta r^{(n)}_{catch} \right|}{\left| \delta r^{(n)}_{catch} \right| + \left| \delta r^{(n)}_{field} \right|} \tag{2}$$

where $\delta r^{(n)}_{field} = (\delta r^{(n-1)} + \delta r^{(n+1)})/2$. During normal adaptation, $\delta r_{field}$ decreased in size while $\delta r_{catch}$ increased leading to an increase in the learning index. *Learning Indices* were ordered according to their location in the adaptation sequence (i.e., the sequential numbers of the catch trials), and then averaged to yield the mean learning index curve as a function of trial number.

**Measuring the within-session adaptation process.** To verify that learning converged not only on average, but also in individual sessions, we correlated the motor error (hand deviation) with the trial number. The underlying assumption was that in a case of a monotonic decrease in error along the adaptation session the correlation coefficient should obtain a negative value. On the other hand, low (near-zero) correlation values will correspond to variable adaptation processes with a high trial-to-trial error variability.

**Calculating the error sensitivity and retention factor.** We used a simple linear state-space model (SSM)[38] to model behavioral

adaptation across trials. We assumed that the motor response at trial $n$, which we denote by $r^{(n)}$ followed a simple linear evolution given by

$$r^{(n)} = ar^{(n-1)} + b^{(n)}e^{(n-1)} + \xi^{(n)}. \qquad (3)$$

where $a$ is the retention factor, which measures the exponential decay in the internal state in the absence of error signal, $b^{(n)}$ is the error sensitivity, which could potentially change from trial to trial[38], and $e^{(n)}$ is the error on that trial. The noise term $\xi^{(n)}$ is a random Gaussian variable denoting trial-to-trial noise.

The error in each trial was expressed as the difference between the external perturbation, $\delta^{(n)}$ and the motor output $e^{(n)} = \delta^{(n)} - r^{(n)}$. In our setting, all perturbations were uniform in size, so we scaled the parameters accordingly and defined

$$e^{(n)} = 1 - r^{(n)} \qquad (4)$$

We used the SSM to calculate the retention factor $a$ and the error sensitivity $b^{(n)}$ from our data. The retention factor was defined as the exponential decay of the motor behavior in the absence of error feedback and perturbations. For that purpose, we used washout trials, and calculated the average decay rate as

$$a = \mathbb{E}\left[\frac{1}{P}\sum_m^P \left(\frac{r^{(m)}}{r^{(1)}}\right)^{\frac{1}{m-1}}\right], \qquad (5)$$

where $m$ is the trial index in the washout period with a total of $P$ washout trials, and the expectation, $\mathbb{E}[\cdot]$, denotes averaging across all sessions. Note that there was no significant difference in the retention factor across sessions.

Using the retention factor, we calculated the average error sensitivity in each session in the following manner:

$$b = \frac{1}{N}\sum_n^N \frac{r^{(n+1)} - ar^{(n)}}{e^{(n)}}, \qquad (6)$$

Where $n$ is the trial index. Note that the effects of the noise $\xi^{(n)}$ were neglected since the trials were averaged. To compare the effect of HFS on error sensitivity we used the mean error sensitivity across adaptation trials.

To verify that our model captured the variability in the data, we quantified the goodness of fit between the actual data and our state-space model (see Supplementary Fig. S8). We fit the data to the model using the retention factor and error sensitivity calculated using Eqs. (4) and (5), respectively and calculated the $R^2$ and Variance Accounted For (VAF) for the FF and FF-HFS conditions (FF: $R^2 = 0.84$, VAF = 87.2%; FF-HFS: $R^2 = 0.74$, VAF = 78.0). These results suggest that the retention factor and error sensitivity obtained in our analysis represent a genuine change in the learning dynamics.

**Motor noise and adaptive response.** Motor noise was defined as the mean of the absolute deviations (MAD) across all trajectory errors (measured at maximal deviation) during a set of trials. In each sub-session, noise levels were calculated separately for the control and HFS trials. To match the levels of noise between the control and HFS conditions we used the ratio of the calculated noise values between these conditions and considered those sessions in which the ratio ranged from 0.6 to 1.4 as matched.

To identify the type of motor noise in each condition we calculated the difference between the adaptive response in trial $n$, denoted by $R^{(n)}$, to the external noise. As in previous studies, we defined the adaptive response as the difference between the preceding and the subsequent errors:

$$R^{(n)} = e^{(n+1)} - e^{(n-1)} = r^{(n-1)} - r^{(n+1)} \qquad (7)$$

**Quantifying generalization behavior.** For each of the probe targets (located ±45° from the learned target) and each generalization trial, we first calculated the maximal deviation of the trajectory from the shortest (straight) line connecting the start and end points (generalization points in Fig. 5i, j). Next, we calculated the absolute error (i.e., distance) between these deviations and a two-exponential model (in the form of $f(n) = a_1 e^{-b_1 \cdot n} + a_2 \cdot e^{-b_2 \cdot n}$) best fitted (i.e., minimizing the mean-squared error) to the average adaptation curve calculated across all generalization sessions (solid lines in Fig. 5i, j). To assess the generalization performances along the adaptation session we defined early (trials 1−5) and late (trial number 10th and above) phases of adaptation.

### Neural data analysis

To analyze adaptation, we collected a total of 514 cells from monkey P and 366 cells from monkey S. In addition, to compare the effect of HFS on the 8-target reaching task, we collected 355 cells from monkey C, 239 cells from monkey M and 183 cells from monkey P (on top of the 514 cells used for studying FF adaptation). Different analyses required different selection criteria and thus used different numbers of cells.

**Pre-processing of the neuronal data.** We selected neurons recorded in at least *tr_thresh* trials in each of the four task conditions (see Table 2 for the exact value of *tr_thresh* set for the different analyses). For the selected neurons, spike trains were convolved with a Gaussian kernel of $\sigma$ ms (Table 2). Next, we averaged over trials with similar conditions and formed an order-3 tensor [neurons x conditions x time], which we denoted $X$. Finally, we selected a specific *time window* and z-scored the data over both the condition and time dimensions,

$$X_{z-scored} = \frac{X - \langle X \rangle_{Conditions, TimeWindow}}{std_{Conditions, TimeWindow}(X)}. \qquad (8)$$

### Table 2 | Selected parameters used for each analysis and corresponding figures

|  | Conditions | tr_thresh | σ(ms) | time window (ms) |
|---|---|---|---|---|
| Fig. 3d, e | 8 (eight targets) | 1 | 100 | [−300;0] r.t. GO |
| Fig. 3f–h | 16 (eight targets/±HFS) | 1 | 100 | [−100; +50] r.t. GO |
| Fig. 4a | 4 (±FF/±HFS) | 10 | 100 | [−300;0] r.t. GO (left panel) and [100;400] r.t. GO (right panel) |
| Fig. 4b | 4 (±FF/±HFS) | 10 | 50 | [−500;−400], [−300;−200], [−50;50] and [200,300] r.t. GO |
| Fig. 4d–f | 8 (eight targets) | 1 | 100 | 10 ms |
| Fig. 5b | 8 (eight targets) | 1[a] | 100 | from left to right, [0;5] r.t. CUE; [400;405] r.t. CUE; [−50;−45] r.t. GO; [50;55] r.t. MO |
| Fig. 5c | 8 (eight targets) | 1[b] | 100 | [−200;0] r.t. GO |

[a]Trial selection is described below in PCA dimensionality.
[b]The trial selection criteria for the CCPA analysis are described in the Cross Condition Generalization Performance (CCGP) section below.

We ran a PCA on the z-scored data by first concatenating conditions and time windows, to generate a matrix calculated on *tr_thresh* with dimensions neurons x [conditions x time_window].

**PCA dimensionality.** To estimate the dimensionality of the neural representation, we calculated the participation dimension of the mean activity pattern across all targets. We first selected neurons with more than 40 trials across all targets under HFS (in addition to the *tr_thresh* condition on trials per target as described above). We then averaged the trials for each condition, resulting in eight mean vectors for each ±HFS. For the two conditions, we computed the correlation matrix across the 8 targets, and calculated the participation dimension,

$$D_{PCA} = \frac{(\sum_n \lambda_n)^2}{(\sum_n \lambda_n^2)} \qquad (9)$$

Here, $\lambda_n$ for $n = 1, \ldots, 8$ stand for the eigenvalues of the correlation matrix targets. averaged across all trials for a specific condition. Note that we subsampled trials after grouping the different targets so that both the Control and HFS data tensors would have the same number of trials for a given target, and averaged across different trial sections. As a result, the variance for control was higher due to the greater number of trials.

**Null Field/FF classification accuracy.** The data were preprocessed with parameters **tr_thresh** = 10, **σ** = 50 ms, **time window** = 100 ms and the four task conditions (±FF/ ± HFS). Individual trials were projected onto the first 15 PCs and a test set was constructed by taking half of the total number of trials. A linear classifier was trained using the LinearSVC library on the training set composed of the remaining trials to discriminate between Null Field and Force Field under both Control and HFS conditions. We computed the classifying accuracy for 10,000 samples of test and train sets (Fig. 4c) by taking the rate of correct classifications over both the number of test trials for a given sample and the time window.

**Targeted dimensionality reduction (TDR).** We applied a targeted dimensionality reduction analysis[39] to our dataset to associate variations in neural activity to the position of the cued target. We first preprocessed the data (see above) using the following parameters: *tr_thresh* = 1, $\sigma$ = 100 ms and *time window* = 10 ms. We then expressed the z-scored responses of neuron $i$ at time $t$ to target $k$ (where $k = 1, \ldots, 8$ is the target index) denoted by $s_{i,t}^k$ as a linear combination of task variables; i.e., the $x$ and $y$ positions of the eight cued targets:

$$r_{i,t}(k) = \beta_{i,t}^1 x_{position}^{(k)} + \beta_{i,t}^2 y_{position}^{(k)} + \beta_{i,t}^3. \qquad (10)$$

We then selected the times $t_{max}^{1,2,3}$ at which $\beta_{i,t_{max}}^{1,2,3}$ had a maximal norm over the given time window. Then, we orthogonalized the set of vectors $\beta_{i,t_{max}}^{1,2,3}$ with the QR-decomposition to derive a set of three TDR axes; the first two define a plane that best explained the $x$ and $y$ positions of the cued target and the third was a fixed bias axis.

**Neural angle analysis.** To calculate the neural angle, we only considered neurons with minimal numbers of trials (-FF: 15 trials for monkey S and 10 for monkey P; +FF: 25 trials). Next, we computed the target-related TDR axes using the neural data for the eight targets under the control conditions. We trial averaged the neural activity for different HFS/FF conditions, and projected them on the target plane found using the TDR analysis. The points resulting from the projections of the neural states were parameterized in polar coordinates by their radius and angle (r,θ). Thus, the angle θ obtained during movement preparation predicted the chosen target while the radius r indicated the strength of this prediction. We eliminated projections that

fell below a threshold radius ratio **rr** calculated as a neural state's radius to the average radius of the neural states used for computing the TDR axes,

$$rr = \frac{r}{\langle r_{Control8} \rangle} > 0.5 \qquad (11)$$

The selected angles corresponding to 10 ms time windows in the range of −500 to −150 ms around the GO signal for monkey S (−200 to 0 around the GO for monkey P) were aggregated to generate angular histograms.

**Cross-condition generalization performance (CCGP).** We evaluated the cross-condition generalization performance (CCGP) of the target-related neural manifold, as described by ref. [42] separately in -HFS/-FF and +HFS/-FF conditions. We first selected neurons with more than 40 trials, all targets considered, in the -HFS/-FF and +HFS/-FF conditions. Next, we selected trials to ensure an equal number of trials for all neuron/target combinations between -HFS/-FF and +HFS/-FF. Then, we applied the preprocessing procedure with the parameters specified in the table, and created 1000 neural trajectories by random trial index shuffling for all neuron/target combinations. Finally, we analyzed the 36 pairs of dichotomies available for the targets positioned in a rectangle. For each pair, we determined the hyperplane for the first dichotomy and assessed the performance of this hyperplane on the other dichotomy (using the LinearSVC library). For example, we trained a linear decoder to discriminate between neural activity associated with a top-left and a top-right target, and then tested this decoder on neural activity associated with bottom-left and bottom-right targets. For presentation purposes, we only used dichotomies with CCGP > 0.7 under the -HFS/-FF condition (where CCGP = 0.5 is the chance level).

## Computational model

To study the capacity of cerebellar input to reduce cortical dimensionality and improve the generalization of the linear readout we implemented a computational model of cerebellar-to-cortical interactions. The model consisted of a simple recurrent cortical layer and a low-dimensional feedback loop[71,72] where the cortex receives an external command indicating which out of the eight possible targets was selected. Importantly, the command was unstructured and did not contain any topological relations between targets. The cerebellar feedback was trained to read out the cortical activity and return the coordinates of the selected target on a 2- dimensional plane (structured input).

The dynamics of the cortical neurons are given by the circuit equations[73]:

$$\tau \frac{d}{dt} x(t) = -x(t) + \frac{g}{\sqrt{N}} W^{ctx} r(t) + \alpha W^{ex} u^0 + \beta W^{crb} z(t). \qquad (12)$$

where $x(t)$ is a $N$-dimensional vector representing the membrane potential of the cortical neurons. The instantaneous firing rate of each neuron $i$ is given by $r_i(t) = \tanh x_i(t)$. The dense cortical recurrent connectivity is given by the $N$-by-$N$ matrix $W^{ctx}$, whose entries are sampled, *i.i.d.* from a standard distribution (a normal distribution with zero mean and unit variance). The parameter $g$ measures the disorder in the system and provides the network with cortex-like activity [Sompolinsky et al., 1988, Kadmon and Sompolinsky, 2015]. The 8-dimensional vector $u^0$ is a 1-hot encoded vector representing the target identity: for each target, one component of $u^0$ is '*1*' while the rest are '*0*'. The external input is fed into the network through the $N$-by-*8* matrix $W^{crb}$, whose entries are sampled *i.i.d.* from a standard distribution. The last term represents the cerebellar feedback. Here $z(t)$ is a 2-dimensional vector encoding the coordinates of the target on

the target plane. We followed previous studies[71,74–77] and modeled the cerebellar feedback as a simple linear transformation of the cortical activity

$$z(t) = W^{ro}r(t). \tag{13}$$

where the 2-by-$N$ cortical readout matrix $W^{ro}$ is trained to produce the correct target in $z$ using linear regression in an open loop model (i.e., setting $z$ to the correct values in Eq. (13) and learning $W^{ro}$ through linear regression from the mean activity of ($rt$) in for each target. The model has three parameters: the amount of disorder $g$, the efficacy of the unstructured input $\alpha$, and the efficacy of the structured feedback $\beta$. Cerebellar blocking was simulated by reducing the efficacy of the structured feedback $\beta$. We trained the readout $W^{ro}$ for each value of $\beta$, indicating adaptation to the cerebellar blockage. For each value, we calculated the PCA dimension (participation ratio) and CCGP of the cortical activity using the full cortical population. In the model, $\beta$ remained finite even during "HFS" trials, thus corresponding to the task structure learned by the cortex through other connected areas (e.g., the Basal Ganglia), or incomplete blockage of the cerebellar output. We repeated the simulation 600 times. In each simulation we randomly samples the cortical connectivity ($W^{ctx}$), the cerebellar and external input ($W^{crb}$ and $W^{ex}$), and learned the appropriate readout matrix ($W^{ro}$).

## Statistical analyses

All the behavioral data and single cell analyses were conducted using MATLAB software (R2022a by Mathworks). Differences between adaptation in the control and HFS conditions were assessed using two-way ANOVAs, with learning bins as one factor and the presence or absence of HFS as the other factor. Comparisons of the correlations between deviations and trial number were made using a Wilcoxon rank sum test. Comparisons of learning indices (LIs) in catch trials and error sensitivity at different noise levels implemented two-way ANOVAs. Comparisons of generalization across stimulation conditions was assessed by two-way ANOVA. In all comparisons, the significance level was set at 0.05.

## Reporting summary

Further information on research design is available in the Nature Portfolio Reporting Summary linked to this article.

## Data availability

Source data are provided as a Source Data file. The data that support the findings of this study are available from the corresponding author (Y.P.) upon request. Source data are provided with this paper.

## Code availability

The code used for the analysis is available at https://github.com/kadmon-lab/cortico-cerebellar-pathway (https://doi.org/10.5281/zenodo.14778754).

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

## Acknowledgements

This work was funded by the Israel Science Foundation (ISF-1207/23 to Y.P., ISF-1821/23 to J.K. and ISF-2484/23 for F.M.), the Deutsche Forschungsgemeinschaft (431549029-SFB 1451 to Y.P.), the BSF (2023321 for Y.P. and 2021323 for F.M.), the National Institute Of Neurological Disorders And Stroke of the National Institutes of Health under Award Number R01NS110901and the generous support of the Baruch Foundation (to Y.P.) and the Azrieli Foundation (to J.K.).

## Author contributions

Y.P. conceived the project. S.I. and R.H. ran the experiments. S.I. collected the data. S.I., H.N., and O.R. analyzed the data. L.E. simulated the computational model. F.M., J.K., and Y.P. supervised the project. Y.P., J.K., F.M., H.N., and S.I. wrote the original draft of the paper.

## Competing interests

The authors declare no competing interests.
