## [Transparent Peer Review file · Nature Communications]

Cerebellar output shapes cortical preparatory activity during motor adaptation

Corresponding Author: Professor Yifat Prut

Version 0:

Reviewer comments:

Reviewer #1

(Remarks to the Author)

In the present manuscript, the authors studied effects of cerebellum on motor learning and cortical preparatory activity during force-field adaptation. By blocking cerebellar output using high-frequency electrical stimulation, they found that both motor stability and adaptivity were impaired. At neural level, preparatory neural states of motor cortex became more complex and less generalizable, which was also replicated in a simplified computational model. Dealing with the adaptation-related impairments, which was mainly attributed to the reduce in error sensitivity, the motor cortex exhibited FF-related rotation in the preparatory state as a compensation strategy. Overall, the elaborated experimental design and comprehensive data presentation make the manuscript appropriate for publication in Nature Communications.

Here are some specific issues to be addressed:

1. SCP stimulation. In this work, cerebellar outflow was blocked by applying HFS on SCP. This manipulation led to impaired adaptation and increased motor variability. Significant changes in neural state of motor cortex were also observed, especially during motor preparation. However, the cerebellar projections to other areas such as premotor, PPC, and basal ganglia, which were also probably involved in this task and disturbed during the stimulation. Thus, the effect of HFS might be complicated. Some behavioral results, like internal noise, might be attributed to changes in other areas in addition to M1. Moreover, the alteration of neural geometry in motor cortex could not be affirmed entirely as the direct consequence of cerebellar block. At least, this issue should be discussed, in combination with neural circuit or MRI studies.

2. Task.

1) Sequence of conditions composing one session was shown in Fig.1b: Control, FF, WO, HFS, WO, FF-HFS, WO. Was the sequence constant across all sessions? If so, it should be considered and discussed whether the non-random order of FF, HFS, FF-HFS conditions would be memorized by the well-trained monkeys.

2) For FF adaptation task, the temporal parameter settings of pre-cue, delay, reach for two monkeys were different (Line 451~455). Is there any consideration for that? Is there difference in behavioral performance between the monkeys? Could the authors provide behavioral statistics for each monkey in the supplemental materials?

3. Bizzi lab (Padoa-Schioppa et al., 2002, Neuron) reported that cortical activity reflects the movement dynamics during the preparatory period in the force-field task, which is highly relevant and should be acknowledged and discussed, compared with the present study.

4. Have the behavioral results and neural activity in washout conditions ever been analyzed, especially in the early washout? It might provide additional insight into learning effect.

5. The poor adaptation under HFS was measured using correlation coefficients in Fig.1e. Would it be more appropriate to demonstrate using goodness-of-fit, as the curves in Fig.1c were seemingly close to exponential?

6. Fig.1g, the line of FF-HFS was shorter than that of FF. Were the trial numbers of these two conditions different? It should be clarified.

7. Adaptive response (Fig.2g-h).

- 1) The definition of adaptive response was the difference between deviation in trial n+1 and trial n-1. Why not calculate the difference between adjacent trials (trial n+1 and trial n)?
 - 2) Fig.2h, although the statistical result showed that FF produced a significant adaptive response, the right point (trials with larger positive error) seems to contribute most. Would the significant difference also exist if these trials were removed?
8. PCA.
- 1) Population dimensionality reduction analysis showed the task-related activity and the effects of HFS on preparatory neural states (Fig.3d-h). The explained variance of each PC should be showed.
 - 2) As the PC1-3,5 showed both the target-dependent and -independent effects of HFS (Fig.3f-h). How about the conditions in PC4?
9. Fig.4c, violin plots might be better to show the distribution of accuracy values during each period.
10. Computational model. Since the model could simulate the cerebellar regulation of motor cortex and captured the effects (higher Dpca) of cerebellar block, is it possible to simulate and explain the transformation from "high Dpca before CUE" to "low Dpca before GO"? Have the authors ever tried to simulate the cortical activity after GO during trials with HFS, and get a low Dpca (as a result of compensatory shift in motor cortical activity)?

Minor points:

1. For all the figures with population analysis on neural activity or behavioral sessions, n number should be labelled.
2. Fig.1 legend, the two subgraphs in Fig.1d should be labelled as "top" and "bottom", instead of "left" and "right", respectively.
3. Fig.3 legend, the serial letters (b-g) for subplots should be bold.
4. Fig 4 legend, "-300 ms to 0 around the Go signal" in line 5 could be replaced with "-300 ms to 0 before the Go signal".
5. Looks like the "TO+50" should be "GO+50".

Reviewer #2

(Remarks to the Author)

In this excellent manuscript from the Prut Lab, the authors study the role of the cerebellum and the motor cortex in the process of learning of arm movements. They convincingly show that when cerebellar outputs to the cortex are disrupted, nearly all behavioral signatures of reach adaptation are negatively impacted. They then focus on the neural activity in the motor cortex and uncover the remarkable finding that in response to the disruptions of the cerebellar output, the cortex is attempting to compensate via re-aiming. The concept of re-aiming has been hinted at in the behavioral literature but not before examined using neurophysiological approaches.

To help improve the work, I have the following main suggestions:

Although the focus of the behavioral analyses is on measures of adaptation, it would be better to also show whether reach characteristics, such as reaction time, and the ability to stop the movement on target, were affected by the stimulation. Some of the current work in mice from the Person Lab suggests that the cerebellum's output is critical for stopping the arm on target. Is this ability disrupted in the monkeys? Moreover, other works in mice shows that the preparatory activity in the motor cortex depends on cerebellar output, and the disruption of that preparatory activity affects the choice that the animal is about to make. Thus, I wonder whether HFS here altered any aspects of reach onset or kinematics.

The TRD analysis and the finding that during FF condition the neural activity was centered around the target but then rotated under the HFS condition were fascinating. A useful control would be to apply the same analysis during NF conditions, where we would now not expect to see rotation under the HFS condition. If indeed the animals are re-aiming, are there kinematic patterns in the reaches that follow from the trial-by-trial neural analysis that can confirm that conjecture? That is, can you use the neural analysis to predict upcoming reaches, particularly in catch trials? What I am suggesting is to look for within trial behavioral consequences in the preparatory neural data.

Here are some minor suggestions:

Fig. 1B. provide number of trials on the x-axis of part b.

Fig. 1C. part c is difficult to understand. What are the various sets? If a single session in a single monkey is being shown, specify that. Provide scale to indicate distance.

Fig. 1F. provide scale

Fig. 1e. Unclear what this is. Wouldn't the slope of the regression be more useful than correlation coefficient? Maybe show some of the actual data?

Fig. 3b. Y-axis scale on Fig. 3B is incorrect.

Fig. 3f. Part f is in a sense trivial because you already show that the level of activity in the cells is much lower in HFS trials, and thus the separation here is merely a reflection of that fact. Maybe just point this out.

Fig. 3a. I could not see asterisks in this plot.

The fact that in the prep period there is a bigger difference between FF and NF during HFS than no HFS is interesting. It would be useful if you could show examples of this.

Line 251. If this interpretation is correct, then you should see some evidence for it in the initial part of the reach trajectory, pulling the arm away from the target, particularly in catch trials, especially if you could do this in trials in which the neural activity during the prep period indicates reaiming.

Line 259. But in mice decision-making tasks we know that cerebellar output to M1 is essential for maintaining the ramping activity during the delay period.

Line 260. The way to test this is to show that when HFS is applied during NF condition there is no rotation.

Line 263. Show the firing rates during null field and FF HFS conditions.

With my best wishes, --Reza Shadmehr

Reviewer #3

(Remarks to the Author)

Israely, Ninou et al report on the role of cerebellum in adapting to force fields. They use high-frequency SCP stimulation ("HFS") to disrupt cerebellar outputs, and record behavior and M1 neurons (and maybe PM) during FF adaptation. Their central claims are that HFS impairs behavioral adaptation consistent with lowering error sensitivity; HFS prevents normal adaptation in M1 and instead causes activity to look like re-aiming; HFS raises representational dimensionality in M1; HFS impedes generalization of learning; and a low-D cerebellar feedback model reproduces these results. This is a fairly interesting set of results, but I have concerns about several central analyses that make it difficult to assess the reliability of the findings.

The core strength of this paper, though, is that there is little prior work (to my knowledge) on the impact of cerebellar outputs on cortical motor structures. Disrupting this pathway and examining motor cortices is important to understanding how cerebellum plays its role in primates. I believe that at least of few of the key results are solid.

Major concerns

1. I could not find details about the recordings, and results were not broken down by area. I found statements that recordings were made from "primary and premotor cortical areas", and that the recordings were "mostly in [M1]". Unless I missed the details, this is grossly insufficient to even fully interpret the results let alone reproduce them. Here are some examples of missing information. From where in M1 were recordings made? (Surface or sulcus? What part of the representation?) How were the recording spots chosen or identified? How many cells were recorded in PM? Where in PM? Were these recordings different? Were recordings spike sorted? How? Were multiunits included? Was drift checked for? Were only stable recordings included? Did neural effects of HFS vary by recording location?

2. In the analysis where learning is attributed to reduced error sensitivity or reduced retention, how well does the model fit the data? If the model accounts for nearly all of the learning (after accounting for trial-by-trial noise), then this is a strong and interesting result. But if it's not the right model structure and only some modest fraction of the learning is being fit, attribution of learning to one parameter or the other would not be meaningful.

3. Dimensionality. Determining dimensionality is always tricky in real data, and different measures tell you different things. Here, the authors use the Participation Ratio, which works purely on the eigenvectors – the distribution of variance across dimensions. It does not (cannot) distinguish "real" variance from noise. Adding a real, high-dimensional signal pushes the PR up, but so does adding noise, which is also generally high-dimensional. Adding a low-dimensional signal, either a large one or one that is redundant with information already in the system, pushes the PR down given the exact same noise level. This makes it hard to interpret Figure 5a,c. The authors argue that cerebellar inputs "quench the dimensionality of the cortical manifold". Alternatively, the small drop observed in dimensionality could be due to a reduction in noise at cue onset (see papers by Churchland and Shenoy, Rajan and Abbott, and others); or they could be due to the addition of a real low-D signal from any of a number of inputs, including motor thalamus, on which there is a burgeoning literature. The higher dimensionality during HFS could be due to the loss of a low-D input as the authors suggest, or could be due to the injection of noise, or to reduction of stability in the network for any reason (which we might expect from the increased behavioral variability). Looking at Supplementary Figure 3, the drop in estimated dimensionality seems to occur just after Cue and reach a minimum just before Go. This is a similar time course to the quenching of variability known in M1 and PMd (e.g., Churchland 2010, Nat Neurosci). In sum, it is hard to know which of several sources causes the effects on dimensionality observed here, and interpretation should be cautious.

4. Model. I don't have any specific problem with the model in Figure 5, but I was unclear why it took this structure in particular. If the motivation was solely from the dimensionality results, I explain above why this is not enough to motivate a single model. If there's literature motivating it, please cite it. If it's just a plausibility argument, please make this clearer.

Other

5. The second control for higher trial-by-trial error due to HFS, of categorizing sessions as low or high noise, is unhelpful. The primary control, in which the distributions are matched, is fine. It's not clear to me why the second control would help in any way, since even within noise group you're still comparing higher noise in the FF-HFS condition to lower noise in the FF.

6. Figure 1h: the FF regression line does not appear to fit the data well at all. This may be the best-fit line because of the huge distribution for that first point (i.e., I'm not suggesting this is an outright error), but that fit is far from convincing. Is it just this way of visualizing the data, or is there a problem here? Can the authors justify this model when it fits so poorly?

7. Interpretation of the PCA plots (Figure 3)

7a. PC2 has a large offset between control and HFS, which appears to show that the HFS is having a direct effect on neural firing rates. This is presumably from the tonic activation of the cerebello-cortical pathway. Given that it pushes firing rates away from their typical values, HFS may perturb the circuit's function directly and not solely through preventing normal phasic and structured inputs. This should be mentioned.

7b. The interpretation of Figure 3h is suboptimal. The authors describe the effect of HFS as "inverse" vs. in other PCs. This is literally true, but somewhat misleading. The larger picture is that the pattern of targets is slightly bent in 5-D, and although you can find a dimension in which representation is flipped, it probably doesn't give a good picture of the larger structure.

8. Please choose a different name for your dimensionality measure. "dPCA" is already the name of a method. "D" would be fine.

Minor editing error and typos

Supp Fig 2: is the on-figure legend wrong here? Colors in the on-figure legend are reversed relative to the caption and to Fig 4.

L190: citations not inserted correctly

L231: typo: should be TDR, not TRD

"tr_thresh" written "tr_tresh" in several places

Version 1:

Reviewer comments:

Reviewer #1

(Remarks to the Author)

All of my previous concerns have been adequately addressed.

Reviewer #2

(Remarks to the Author)

Thank you for the revisions. I believe the work is substantially improved and I have no further comments.

Reviewer #3

(Remarks to the Author)

The authors have done an excellent job of addressing the concerns I raised. The manuscript is of clear value to the field, and the results are solid.

Dear Editor, Dear Referees,

Please find enclosed our revised manuscript entitled "**Cerebellar output shapes cortical**
**preparatory activity during motor adaptation**". We would like to thank the reviewers for
their thoughtful feedback and important suggestions. We carefully adhered to all the
recommendations made by the reviewers. All changes in the manuscript are highlighted in
red. Specifically, we made the following changes to the manuscript:

- 1. To respond to the comments from the reviewers, we added a description of the
effect of the cerebellar block and task conditions on movement parameters. We
expanded and quantified the simulation analysis. We added a new supplemental
figure (Supp. Fig. S7) and a supplemental table (Sup Table S1), which includes the
mean trajectory and velocity profiles as well as the parametric analysis of the effect
of FF and HFS on reaction time, movement time, and peak velocity (using 2ways
ANOVA).
- 2. To address the point made by reviewer #3, who suggested that the increase in
dimensionality during HFS conditions could reflect added noise, we analyzed the
effect of HFS on trial-to-trial variability (estimated by the Fano Factor). This point has
now been added to the text and depicted in a supplemental figure (Supp. Figure S6).
- 3. To validate the state-space model that we used to explain error sensitivity in the
control and HFS conditions, we calculated the goodness of fit of the model to the
data in the two different conditions. This point is now depicted in Supplemental
Figure S8.
- 4. We provide a more detailed account of the across-monkey differences in task
parameters (in addition to the across-monkey behavioral comparison) summarized
in the Method section and in a new table (Table I). In our initial submission, we used
monkey-specific parameters to study the learning-related changes in the neural
angle in the presence/absence of cerebellar signals (in Figure 4h). We now added
new panels to Supplemental Figure S4 which present data using the same criteria for
both monkeys
- 5. We added cell numbers and distance scales in all figures.
- 6. Two reviewers suggested that the behavioral analysis of within-session variability
was unclear and might not contribute significantly to the conclusion of impaired
adaptation. We therefore removed this analysis from the current version (both text
and related panels in Figure 1).
- 7. The analysis of adaptive response was re-run using the pooled data, and the single
monkey data are now shown in Supplemental Figure S1.
- 8. We expanded the method section to describe the recording and how single units
were extracted from the raw signal. We also added cortical maps (Supplemental
Figure S2) to present the recording sites in the two monkeys.

We believe that the current version of the manuscript has been greatly improved in clarity
and rigor compared to the original submission and complies with all the requests for
revision made by the reviewers.

Yours sincerely,

Yifat Prut

**Reviewer #1 (Remarks to the Author):**

In the present manuscript, the authors studied effects of cerebellum on motor learning and cortical
preparatory activity during force-field adaptation. By blocking cerebellar output using high-
frequency electrical stimulation, they found that both motor stability and adaptivity were impaired.
At neural level, preparatory neural states of motor cortex became more complex and less
generalizable, which was also replicated in a simplified computational model. Dealing with the
adaptation-related impairments, which was mainly attributed to the reduce in error sensitivity, the
motor cortex exhibited FF-related rotation in the preparatory state as a compensation strategy.
Overall, the elaborated experimental design and comprehensive data presentation make the
manuscript appropriate for publication in Nature Communications.

Here are some specific issues to be addressed:

1. SCP stimulation. In this work, cerebellar outflow was blocked by applying HFS on SCP. This
manipulation led to impaired adaptation and increased motor variability. Significant changes in
neural state of motor cortex were also observed, especially during motor preparation. However, the
cerebellar projections to other areas such as premotor, PPC, and basal ganglia, which were also
probably involved in this task and disturbed during the stimulation. Thus, the effect of HFS might be
complicated. Some behavioral results, like internal noise, might be attributed to changes in other
areas in addition to M1. Moreover, the alteration of neural geometry in motor cortex could not be
affirmed entirely as the direct consequence of cerebellar block. At least, this issue should be
discussed, in combination with neural circuit or MRI studies.

**The reviewer is right in pointing out that the cerebellar output targets several brain sites other than**
**the motor cortex and some of the observed behavioral effects could be related to these interactions.**
**While we acknowledge that the SCP terminates on other subcortical and cortical areas, we focused**
**in this study on motor cortical regions (primary motor and premotor areas, see recording maps now**
**added as Supplemental Figure S2 in the revised manuscript) because these regions are often**
**implicated in motor learning and, importantly, previous studies have shown that premovement**
**changes in motor cortical activity probably drive motor adaptation (Paz, Boraud et al. 2003, Vyas,**
**O'Shea et al. 2020, Sun, O'Shea et al. 2022). Since the motor cortex is a major target for cerebellar**
**efferents, particularly in human and non-human primates(Hore and Flament 1988) we specifically**
**sought to study whether the motor cortical correlates of motor learning are driven by cerebellar**
**efferents. Clearly other areas as well (the PPC for instance) could further contribute to the observed**
**behavioral changes under HFS, but it is not clear whether they also take part in driving the adaptive**
**motor response. This point has been added to the Discussion with the appropriate references (lines**
**371-375).**

2. Task.

1) Sequence of conditions composing one session was shown in Fig.1b: Control, FF, WO, HFS, WO,
FF-HFS, WO. Was the sequence constant across all sessions? If so, it should be considered and
discussed whether the non-random order of FF, HFS, FF-HFS conditions would be memorized by the
well-trained monkeys.

**The reviewer is right and the order of the main epochs (Control8, Control1, HFS, FF, FFHFS) was fixed**
**and therefore could have been predicted and potentially affect its adaptive behavior. We have no**
**direct way to test this. Nonetheless, following this comment, we went back to the data and**

examined whether this predictability may have affected task performance. We found that during a
 single recording day in which there were several learning sessions (with the same block order but
 with different learned targets) there was no trend toward improved performance (see **Figure R1**
 here). Specifically, the learning rate and the residual errors in the second or third learning sessions
 were not significantly different from the first session.

Clearly, we cannot exclude the possibility that across different conditions, the fixed order of blocks
 made the first trial in each block less surprising. However, since in any case all trials in the session
 were directed towards the same single target, we speculate that the added information of the
 transition between blocks did not affect behavior in a substantial manner. However, this issue needs
 to be addressed explicitly. This point has been added to the manuscript (Method section, line 518-
 524).

**Figure R1: Effect of learning sets order on learning curves.** Averaged FF learning curves calculated for
 single learning sets, divided based on set order. On each recording day we normally administered 2-3 learning
 sets. In each set, the conditions (i.e., control, FF, FF-HFS) appeared in a fixed order. We found no significant
 effect of set order on learning performance.

2) For FF adaptation task, the temporal parameter settings of pre-cue, delay, reach for two monkeys
 were different (Line 451~455). Is there any consideration for that? Is there difference in behavioral
 performance between the monkeys? Could the authors provide behavioral statistics for each
 monkey in the supplemental materials?

To address this point we have added a supplemental figure that summarizes the behavioral
 parameters of the two monkeys, including RT and MT and velocity profiles (**supplemental figure SR4**
 **and Supplemental TableR1**)

Task parameters were not identical for the two monkeys for two reasons: (1) The two monkeys
 differed in terms of their sensitivity to the duration of the delay and the application of HFS. We
 therefore had to slightly modify task parameters accordingly. We adjusted the task parameters for
 purposes of comparison (in terms of number of trials and success rate) across monkeys. (2) Based on
 the experience gained from the first monkey, we slightly changed the duration of epochs to optimize

the number of trials obtained in the different conditions with sufficiently long epoch duration. Please
note that in each analysis we compared the single-monkey results to ensure consistent outcomes
despite the differences in task parameters.

3. Bizzi lab (Padoa-Schioppa et al., 2002, Neuron) reported that cortical activity reflects the
movement dynamics during the preparatory period in the force-field task, which is highly relevant
and should be acknowledged and discussed, compared with the present study.

We thank the reviewer for this highly relevant reference, which we think supports our findings. In
this study the authors looked for cortical correlates of the kinematic-to-dynamic transformation
during the pre-Go preparatory period. The authors focused on single-cell properties and identified a
significant shift in the preferred direction (PD) of SMA neurons during the pre-Go period that was
correlated with the adaptation behavior of the animals. Interestingly they found a substantially
weaker effect in the premotor areas and no effect in M1. This is similar to our finding of no shift in
neural angle when considering the entire recorded population of motor and premotor cells (though
our population was mostly composed of M1 neurons). It is possible that the compensatory shift we
observed in motor cortex when cerebellar signals were blocked is at least partially driven by SMA-to-
M1 inputs. This is particularly interesting since the SMA in itself is less likely to be affected by the
cerebellar output block. Verifying or refuting this hypothesis would require studying the SMA under
HFS. We added this reference to the introduction (line 57) and further explore this point in the
Discussion section (line 397-401).

4. Have the behavioral results and neural activity in washout conditions ever been analyzed,
especially in the early washout? It might provide additional insight into learning effect.

We analyzed the behavior of the monkey during washout trials to calculate the retention factor used
for quantifying the learning process. Therefore, the washout trials were fundamental for quantifying
the adaptive behavior of the different conditions.

5. The poor adaptation under HFS was measured using correlation coefficients in Fig.1e. Would it be
more appropriate to demonstrate using goodness-of-fit, as the curves in Fig.1c were seemingly close
to exponential?

Following the reviewer's comment (also made by Reviewer #2) we felt that this analysis added little
to the manuscript and was probably better captured by the significant decrease in error sensitivity
we already reported. We therefore decided to remove this analysis from the manuscript.

6. Fig.1g, the line of FF-HFS was shorter than that of FF. Were the trial numbers of these two
conditions different? It should be clarified.

The reviewer is right: there were some differences between the two monkeys. In monkey P, the
sequence of FF and FF-HFS were the same length (50 trials) but for monkey S we used 70 trials in FF
but only 50 trials in FF-HFS. This point has now been clarified in the results and method sections in
the manuscript (lines 511-516) and the figure caption was modified to explain this point.

7. Adaptive response (Fig.2g-h).

1) The definition of adaptive response was the difference between deviation in trial $n+1$ and trial $n-1$.
Why not calculate the difference between adjacent trials (trial $n+1$ and trial n)?

2) Fig.2h, although the statistical result showed that FF produced a significant adaptive response, the
right point (trials with larger positive error) seems to contribute most. Would the significant
difference also exist if these trials were removed?

7.1. The definition of adaptive response was taken from Ranjan and Smith 2018
 (<https://scholar.google.com/scholar?cluster=5684349341964854049&hl=en&oi=scholarsee>).
 Drawing on this work, we tested the relations between the adaptive response calculated based on
 the error on flanking trials ($e_{n-1}-e_{n+1}$) and the error on trial n (e_n). Using e_n in both the numerator and
 the dominator would yield an artifactual correlation.

We used this analysis to test the possibility that HFS in itself acts as an external source of noise,
 which was shown to reduce error sensitivity (Albert, Jang et al. 2021) and impair adaptation. In our
 study this point is important because it lends weight to our claim that impaired learning under HFS is
 a primary effect driven by the cerebellar block (similar to the observed deficits in cerebellar patients)
 and not a simple outcome of increased motor noise.

7.2. Inadvertently, the data shown in Figure 2h in the paper present the relationship between the
 adaptive response and the error as calculated for only one monkey. We now show this result for the
 monkeys separately (Figure R2-A-C) as well as the combined data for the two monkeys (Figure R2-D).
 The results were not identical across the two monkeys, but the trend in both cases was similar and
 the significance tests revealed similar results. In the manuscript we replaced panels g and h in Figure
 2 with the pooled data (shown here in Fig. R2-D) and presented the single monkey data in
 Supplemental Figure R1.

**Figure R2: Relations between adaptive response and error in different trial conditions.** Monkey-specific
 adaptive responses were calculated for each monkey during the Control conditions (A), HFS (B) and FF (C).
 The regression slope is shown for each graph. D Mean adaptive response calculated by pooling the data from
 both monkeys. Each r value was tested for a significance difference from zero slope (t -test). Differences in
 regression slopes were tested and found to be non-significant when comparing the Control to the HFS
 conditions, but statistically significant when comparing the Control to the HFS conditions.

8. PCA.

1) Population dimensionality reduction analysis showed the task-related activity and the effects of
 HFS on preparatory neural states (Fig.3d-h). **The explained variance of each PC should be shown.**

2) As the PC1-3,5 showed both the target-dependent and -independent effects of HFS (Fig.3f-h).
 How about the conditions in PC4?

8.1 Following the reviewer's suggestion we have added the explained variance to the figure caption
 for both panels d-e (PC1:0.32; PC2: 0.17; PC3: 0.11; PC4: 0.08; PC5: 0.06) and panels f-h (PC1: 0.2,
 PC2: 0.16, PC3: 0.10, PC4: 0.07; PC5: 0.05).

8.2 Following the reviewer's comment we show the plot for PC4 vs. PC1 (Figure R3). In general, we
 found two kinds of effects of HFS on the PCA components: target-independent (e.g., for PC2, 3 and

4) and target-dependent (e.g., PC5). A target-dependent effect of HFS means that cerebellar input
 has information about the different targets as early as the preparatory period. Since the effect of
 PC4 was target independent (as shown for PC2 and 3) we decided not to present it in the
 manuscript. On the other hand, since PC5 was the first PC to present a target-dependent effect of
 HFS, we included it in the manuscript. We now mention this point in the manuscript, line 203.

**Figure R3: Neural trajectories projected on the plane spanned by PC4 x PC1.** Activity was projected on PC1
 and PC4. Solid lines represent neural trajectories in the Control conditions and dashed lines represent the
 neural trajectories in the HFS conditions. In this case, the effect of HFS was target-independent as can be seen
 in the target-specific alignment between Control trials (solid lines) and HFS trials (dashed lines).

9. Fig.4c, violin plots might be better to show the distribution of accuracy values during each period.

**Figure 4c (and supplement figure S1) were changed to the violin format.**

10. Computational model. Since the model could simulate the cerebellar regulation of motor cortex
 and captured the effects (higher Dpca) of cerebellar block, is it possible to simulate and explain the
 transformation from “high Dpca before CUE” to “low Dpca before GO”? Have the authors ever tried
 to simulate the cortical activity after GO during trials with HFS, and get a low Dpca (as a result of
 compensatory shift in motor cortical activity)?

**We appreciate the reviewer’s insightful suggestion regarding the cerebellar modulation of cortical
 activity over time. The data indeed suggest that the cerebellar impact on cortical dimensionality
 evolves over the course of a trial, as evidenced by the changes observed across various stages—trial
 onset, CUE signal, GO signal, and movement—between the Control conditions and the cerebellar
 block conditions.**

**Our current focus was on the preparatory activity, specifically the final cortical representation before
 movement initiation, which we interpret as the initial condition for subsequent movement
 dynamics(Shenoy, Kaufman et al. 2011). Consequently, our model was designed to capture this
 aspect of the cortico-cerebellar interaction and does not account for temporal modulation across
 different trial phases. Addressing the transition from “high Dpca before CUE” to “low Dpca before
 GO” would require a more sophisticated model of cerebellar processing.**

**We agree that a comprehensive understanding of the time-resolved modulation of cortico-cerebellar
 interactions would necessitate incorporating additional features into the model. In particular, this
 would likely involve the inclusion of an external ‘context’ signal to represent the temporal stage**

within the trial and a more nuanced cerebellar processing mechanism, potentially through a neural
layer capable of nonlinearly modulating cortical activity based on context. While such an extension is
beyond the scope of the present work, our study provides an essential first step by demonstrating
how the cortico-cerebellar loop influences cortical dimensionality during motor preparation.

Minor points:

1. For all the figures with population analysis on neural activity or behavioral sessions, n number
should be labelled.

N values were added to all figures.

2. Fig.1 legend, the two subgraphs in Fig.1d should be labelled as “top” and “bottom”, instead of
“left” and “right”, respectively.

The organization of the panels in figure 1 was changed and the location of the subplots in Fig. 1d was
arranged to fit the figure captions.

3. Fig.3 legend, the serial letters (b-g) for subplots should be bold.

Figure caption was corrected.

4. Fig 4 legend, “-300 ms to 0 around the Go signal” in line 5 could be replaced with “-300 ms to 0
before the Go signal”.

Figure caption was corrected.

5. Looks like the “TO+50” should be “GO+50”.

In figure 5A we used the Go Signal to identify the preparatory-related Dpca (using the GO-50 ms
time window). However, to identify the effect of movement on Dpca we aligned the data on the
movement onset event (termed TO) and used a time window of TO+50 to calculate the Dpca after
movement onset. This was done to eliminate the influence of the GO-to-TO timing variability (i.e.,
the trial-to-trial modulation in reaction time).

Reviewer #2 (Remarks to the Author):

In this excellent manuscript from the Prut Lab, the authors study the role of the cerebellum and the
motor cortex in the process of learning of arm movements. They convincingly show that when
cerebellar outputs to the cortex are disrupted, nearly all behavioral signatures of reach adaptation
are negatively impacted. They then focus on the neural activity in the motor cortex and uncover the
remarkable finding that in response to the disruptions of the cerebellar output, the cortex is
attempting to compensate via re-aiming. The concept of re-aiming has been hinted at in the
behavioral literature but not before examined using neurophysiological approaches.

We thank the reviewer for his kind and supportive view of this work.

To help improve the work, I have the following main suggestions:

Point 1. Although the focus of the behavioral analyses is on measures of adaptation, it would be
better to also show whether reach characteristics, such as reaction time, and the ability to stop the

movement on target, were affected by the stimulation. Some of the current work in mice from the
 Person Lab suggests that the cerebellum’s output is critical for stopping the arm on target. Is this
 ability disrupted in the monkeys?

We thank the reviewer for this suggestion and have added a new supplemental figure (supplemental
 Figure S4, supplemental Table R1) and a supplemental table (shown here as well as Figure R4 and
 Table R1) which contain a detailed analysis of the monkeys’ behavior, including reaction time (RT),
 movement time (MT) and maximal velocity. The averaged trajectories and velocity profiles are
 shown together with the mean RT, MT and maximal velocity. A separate table provides the results of
 a two- way ANOVA that was carried out for each of the parameters as well as the interactions
 between the tested conditions.

The question about movement stop behavior is interesting, but our current task design is not
 suitable to address this point. In our task, there is no explicit need for the monkey to stop at the
 target position. In fact, as soon as the monkey moved the cursor through the target, the target
 disappeared and a reward was delivered. This was possible since the monkey had to keep the cursor
 on the target for a brief hold time compared to the size of the target. In this condition the monkey
 was not required to stop its movement for successful completion of the trial. To address the role of
 the cerebellum in stopping the arm at the target, a specifically designed behavioral protocol is
 needed.

**Figure R4: Analysis of behavioral parameters.** The behavioral properties and the effect of HFS were analyzed
 for each monkey (top row - monkey S and bottom row - monkey P). We calculated movement trajectories in the
 different conditions and velocity profiles. We also measured the mean (and SEM) for the reaction time,
 movement time and peak velocities in each of the task conditions. C8 corresponds to the Control condition in
 which the monkeys acquired 1 of 8 targets. C1 corresponds to the Control conditions in which the monkeys
 made a movement towards the one and only target (the Learned Target - LT). In all the remaining conditions
 the monkeys only made movements to the LT. Results of statistical analyses (2-way analysis of variance) are
 shown in Table R1.

Two ways analysis of variance (FFxHFS)

		RT			MT			Peak Vel		
Monkey	Source	DF	F	P	DF	F	P	DF	F	P

P	FF	1	18.84	<< 0.001	1	139.22	<< 0.001	1	65.42	<< 0.001
P	HFS	1	158.76	<< 0.001	1	22.78	<< 0.001	1	57.78	<< 0.001
P	FF*HFS	1	1.97	0.16 ns	1	10.71	0.0011	1	10.51	0.0012
S	FF	1	203.91	<< 0.001	1	198.43	<< 0.001	1	46.72	<< 0.001
S	HFS	1	75.55	<< 0.001	1	361.17	<< 0.001	1	227.91	<< 0.001
S	FF*HFS	1	1.33	0.25 ns	1	4.81	0.028	1	44.12	<< 0.001

**Table R1: Results of 2- way ANOVAs.** The behavioral properties and the effect of HFS on reaction time (RT),
 movement time (MT) and maximal velocity (Peak Vel). For each monkey (P and S) we studied the effect of a
 model of FFxHFS and the possibility of interactions. HFS led to increased RT and MT for both monkeys (with
 no significant interactions) and reduced peak velocity with an interaction effect. Results were similar for both
 monkeys.

Point 2. Moreover, other works in mice shows that the preparatory activity in the motor cortex
 depends on cerebellar output, and the disruption of that preparatory activity affects the choice that
 the animal is about to make. Thus, I wonder whether HFS here altered any aspects of reach onset or
 kinematics.

The reviewer is right and HFS affected other aspects of motor behavior (beyond its effect on
 adaptation). We previously found (Nashef, Cohen et al. 2019) that the HFS affects movement
 kinematics (RT, MT, curvature and shoulder-elbow decomposition) but also the task success rate
 (mostly due to poor timing of movements). In response to the Reviewer's comments, we added a
 summary of motor behavior, including reaction time (RT), movement time (MT) and maximal
 velocity (see figures R4 and table R1 above) during control and HFS trials.

Point 3. The TRD analysis and the finding that during FF condition the neural activity was centered
 around the target but then rotated under the HFS condition were fascinating. A useful control would
 be to apply the same analysis during NF conditions, where we would now not expect to see rotation
 under the HFS condition. If indeed the animals are re-aiming, are there kinematic patterns in the
 reaches that follow from the trial-by-trial neural analysis that can confirm that conjecture? That is,
 can you use the neural analysis to predict upcoming reaches, particularly in catch trials? What I am
 suggesting is to look for within trial behavioral consequences in the preparatory neural data.

The reviewer is right in pointing out that the TDR analysis should be applied to the NF conditions.
 This result is shown in Figure 4G for each of the monkeys. In both cases, the FF-HFS condition was
 rotated relative to the NF condition, but not in the no-HFS (FF) condition.

As for the predictive power of this analysis, we appreciate the reviewer's suggestion and agree that
 demonstrating the ability to predict upcoming reaches based on preparatory activity, both in the
 control and cerebellar block conditions, would be highly impactful. However, the limitations of our
 current dataset prevent us from making reliable single-trial predictions. Specifically, the number of
 neurons concurrently recorded in a single session is insufficient to support such analyses.
 Consequently, our analysis was performed by aggregating neural data across multiple sessions, thus
 allowing us to draw conclusions about population-level neural representations and in particular their
 geometric and topological properties, but not on a trial-by-trial basis.

While this limited our ability to detect the within-trial behavioral consequences in the present data,
 we recognize the importance of this approach and plan to explore it in future work. With more
 advanced neural recording techniques, such as Neuropixels, we expect to be able to address these
 critical questions at the single-trial level, including in catch trials.

Here are some minor suggestions:

Fig. 1B. provide number of trials on the x-axis of part b.

Since the illustration in panel 1b was not made to scale in terms of the number of trials and to
 accommodate the slight differences between the monkeys in the Method sections (line 524) we
 added new Table I that provides the median number of trials per epoch for each monkey (shown
 here as Table R1)

Monkey	Control 8	Control1	HFS	FF	FFHFS	WO
P	64	20	40	50	50	20 (64)
S	64	20	40	70	50	30

Table R1: number of trials per condition per animal

Fig. 1C. part c is difficult to understand. What are the various sets? If a single session in a single
 monkey is being shown, specify that. Provide scale to indicate distance.

The reviewer is right and the explanation of the figure was poor. The panel shows trajectories taken
 from a single session. Since the 4 different conditions (+/- FF, +/- HFS) resulted in different numbers
 of successful trials we divided the trials into 6 equal sets. We now use a color-coded scale for each
 condition and have added the number of trials per condition to the figure itself. We also added a
 scale bar for the distance. We hope this resolves this issue.

Fig. 1F. provide scale

A scale was added.

Fig. 1e. Unclear what this is. Wouldn't the slope of the regression be more useful than correlation
 coefficient? Maybe show some of the actual data?

Following the comment made by the reviewer and a similar comment made by Reviewer #1 we
 decided that this analysis adds little to the paper and is probably not clear. We therefore decided to
 remove it.

Fig. 3b. Y-axis scale on Fig. 3B is incorrect.

The Y- axis was corrected.

Fig. 3f. Part f is in a sense trivial because you already show that the level of activity in the cells is
 much lower in HFS trials, and thus the separation here is merely a reflection of that fact. Maybe just
 point this out.

The reviewer is right: this panel shows a general difference in rate between the Control and HFS. We
 added this panel here to show that a large fraction of the explained variance was related to the

direction-independent difference between the Control and HFS, unlike PC5, which had a direction-
 dependent effect. This point is now explicitly stated in the text (line 204-206).

Fig. 3a. I could not see asterisks in this plot.

We increased the font size of these symbols shown on the blue graph to the right of the raster plot.

The fact that in the prep period there is a bigger difference between FF and NF during HFS than no
 HFS is interesting. It would be useful if you could show examples of this.

On average, the firing rate of cells under HFS was reduced, but the effect of FF on firing rate was not
 necessarily smaller. Below we show three single-cell examples demonstrating this point (**Figure R5**).
 In all cases, activity was aligned on the GO signal (time 0); the application of FF and HFS is indicated
 by the graphs on the right of each raster plot with blue and green curves corresponding to the
 application of HFS and FF respectively. The top curves depict the condition-specific rate function
 (PSTH). Panel A shows an example where movement related activity was in fact higher under HFS
 compared to the Control conditions. Importantly, for all examples, the separation between NF and
 FF before the GO signal was higher under HFS than in the Control conditions.

**Figure R5: Epoch and condition-specific effects on single cell activity.** Three examples of single cells
 recorded during the Control and HFS condition are depicted with and without HFS. All examples are aligned
 on the Go signal and all trials were directed to the Learned Target. Top panel shows the event-related rate
 modulation (PSTH) in the different conditions. To the right of each example the diagram shows the on/off times
 for HFS (blue) and FF (green). The effect of FF on the pre-Go activity in the Control (blue and yellow curves)
 and HFS (orange and purple curves) highlight the effect of HFS on the NF/FF preparatory activity.

**Importantly, the increased NF-FF separation under HFS that we reported (Fig. 4a,b in the**
 **manuscript) was a population-based finding and may not be easy to see in the single cell**
 **examples.** To further address this point and to test whether the effect seen in Figure 4a was merely a
 reflection of global rate modulation under HFS we re-examined the data in this figure and focused
 on the pre-GO (-300 to 0 before the GO event) for both Monkey S (20 trials, 162 neurons) and
 Monkey P (10 trials, 92 neurons). This was done to determine whether the separation in Figure 4a
 was a trivial outcome of the global, condition-specific, change in firing rates. To do so we considered

the 4 centroids in Figure 4a as a n-dimensional vector (n=162 for monkey S and 92 for monkey P) and
 calculated the element-by-element rate changes required for activity to shift between centroids
 (Figure R6 below). The distributions of these changes were centered around zero, indicating that
 the separations between the 4 different conditions cannot be explained in terms of a global shift in
 firing rates.

**Figure R6: Pairwise comparison of the 4 conditions depicted in Figure 4A.** The 4 centroids shown in Figure
 4A were considered as n-dimensional (n was equal to the number of cells available for this analysis in each
 monkey). We tested the element-by-element change in rate required for transitioning between each pair of
 conditions (color-coded histograms). The 4 distributions were centered around zero, indicating that the
 transition between conditions did not correspond to a global change in firing rates.

Line 251. If this interpretation is correct, then you should see some evidence for it in the initial part
 of the reach trajectory, pulling the arm away from the target, particularly in catch trials, especially if
 you could do this in trials in which the neural activity during the prep period indicates reaiming.

We tried to address this comment by calculating the effect of the FF direction (CW vs. CCW) on the
 trajectory deviation during catch trials. For this purpose, we only considered those catch trials which
 were introduced late in the adaptation session (> 10th trial), when we were more likely to find a re-
 aiming strategy. For each adaptation session we calculated the average trajectory deviation during
 catch trials by first normalizing the single session deviations to compensate for the differences in
 velocities in the Control vs. HFS conditions (dividing by the mean deviation on the first two trials) and
 then calculating the mean normalized deviation at the beginning of the movement (130 ms for
 monkey S and 180 ms for monkey P). We used either deviation calculated relative to the center of
 the target. **Figure R8** shows the mean deviation for FF trials (blue bars) in CW and CCW sessions and
 similarly for HFS (red bars). The mean deviations for HFS were larger than in the Control trials, which
 is in line with the idea of re-aiming. However, the noise under HFS was very large, probably due to,
 the increased executional noise which is inherent to this condition and the fact that the number of
 late catch trials per adaptation session was significantly lower under HFS (mean value 2.73 trials)
 compared to the Control condition (mean value 4.09 trials, t-test $p=3.03e-16$). A 2-way Anova (FF-
 direction x Condition) revealed a significant effect for force field direction ($p=0.0034$), but not for
 condition and/or an interaction effect.

**Figure R7: Early trajectory errors in late catch trials.** Mean normalized trajectory deviation calculated at
 early time points relative to the straight line connecting the initial position of the cursor and the center of the
 learned target. Deviations were calculated for catch trials occurring late in the session (after the 10th trial)
 and after separating CW from CCW force fields. Blue bars - FF condition. Red bars FF-HFS conditions.

Line 259. But in mice decision-making tasks we know that cerebellar output to M1 is essential for
 maintaining the ramping activity during the delay period.

The reviewer is right and we have toned down the statement which now reads (line 264-266) “it
 could mean that cerebellar signals are not involved in shaping motor preparatory activity, at least in
 NHPs performing an upper-limb reaching task”.

Line 260. The way to test this is to show that when HFS is applied during NF condition there is no
 rotation.

This issue is shown in Figure 4g, red arrows for each monkey.

Line 263. Show the firing rates during null field and FF HFS conditions.

To address the reviewer’s request below we show the condition-specific NF vs. FF rate distributions
 calculated for the Control (left panels) and HFS conditions. The upper panels show the rate
 distribution calculated in a 500 ms time window before the Go signal and lower panels show the
 firing rate during the same time window after subtracting the condition-specific baseline rate level.

We found that the rate level was generally lower under HFS but that the FF/NF rate distributions
 overlapped in both the Control and HFS conditions.

**Figure R7: Condition specific rate distribution.** Distribution of firing rates calculated for neurons during the
 control (left panels) and HFS (right panels) in the FF and NF conditions. Rates were calculated during the 500
 461 ms before the Go signal. Top row: firing rates. Bottom row: change in rate during the delay relative to the
 462 baseline level. No significant differences were observed in all cases.

With my best wishes, --Reza Shadmehr

Reviewer #3 (Remarks to the Author):

Israely, Ninou et al report on the role of cerebellum in adapting to force fields. They use high-
 frequency SCP stimulation (“HFS”) to disrupt cerebellar outputs, and record behavior and M1
 neurons (and maybe PM) during FF adaptation. Their central claims are that HFS impairs behavioral
 adaptation consistent with lowering error sensitivity; HFS prevents normal adaptation in M1 and
 instead causes activity to look like re-aiming; HFS raises representational dimensionality in M1; HFS
 impedes generalization of learning; and a low-D cerebellar feedback model reproduces these results.
 This is a fairly interesting set of results, but I have concerns about several central analyses that make
 it difficult to assess the reliability of the findings.

The core strength of this paper, though, is that there is little prior work (to my knowledge) on the
 impact of cerebellar outputs on cortical motor structures. Disrupting this pathway and examining
 motor cortices is important to understanding how cerebellum plays its role in primates. I believe that
 at least of few of the key results are solid.

Major concerns

1. I could not find details about the recordings, and results were not broken down by area. I found
 statements that recordings were made from “primary and premotor cortical areas”, and that the

recordings were “mostly in [M1]”. Unless I missed the details, this is grossly insufficient to even fully
interpret the results let alone reproduce them. Here are some examples of missing information.
From where in M1 were recordings made? (Surface or sulcus? What part of the representation?)
How were the recording spots chosen or identified? How many cells were recorded in PM? Where in
PM? Were these recordings different? Were recordings spike sorted? How? Were multiunits
included? Was drift checked for? Were only stable recordings included? Did neural effects of HFS
vary by recording location?

To address this point, we have added more details to the description of the recording technique in
the Methods section (revised section Surgical procedure and recording techniques lines 485-496)
and added recording maps in Supplemental Figure S2.

Specifically, we targeted arm-related areas in the motor cortex. Recording sites were identified
based on distance from the central sulcus and the threshold stimulus level required for producing a
noticeable muscle twitch in the contralateral arm (a 50 ms burst of biphasic stimulation pulses
applied at 300 Hz at an intensity < 70 \$\mu\$ A). We found that of 640 recorded cells, 128 (20%) were
located in high-threshold sites situated > 5 mm anterior to the central sulcus. These premotor areas
were mostly confined to the dorsal premotor cortex. The recording depth was estimated based on
the actual electrode location relative to the depth at which first cortical cells were encountered. The
vast majority of the cells (605 neurons, 94.5%) were recorded from a depth of \$\leq\$ 3 mm, which means
that only a few neurons were recorded from deep in the sulcus. Using offline processing, the neural
signals were filtered (300-6000 Hz), the threshold crossed and upsampled before manually sorting
(using Plexon Spike Sorter). Each neuron was tested for stable activity throughout the recording
session. We included both well-isolated single cells and multiunit activity. Since our single cell
analyses focused on the coordinated activity of neurons and required a large number of neurons, we
pooled the PM and M1 neurons as is commonly done in the field. We also used an additional dataset
from two other monkeys described in our previous papers (Nashef et al 2021, 2022) that was
collected in the same way as described here. The criteria used for cell selections have now been
added to the methods section (lines 551-564).

2. In the analysis where learning is attributed to reduced error sensitivity or reduced retention, how
well does the model fit the data? If the model accounts for nearly all of the learning (after
accounting for trial-by-trial noise), then this is a strong and interesting result. But if it's not the right
model structure and only some modest fraction of the learning is being fit, attribution of learning to
one parameter or the other would not be meaningful.

We thank the reviewer for this insightful comment. As the reviewer suggested, the interpretation of
learning depends on how well the model fits the data (SSM) that were used to estimate the
condition-specific error sensitivity. We calculated two metrics: R-squared and Variance Accounted
For (VAF). For each condition (FF and FF-HFS), we used the retention factor derived from the
washout trials and the average error sensitivity across trials to estimate the model error (**Figure R8**
below). We then compared this estimated error to the actual error (median across sessions) to
compute the R-squared and VAF values. Our results indicate that the model fit the average data
quite well, with R-squared = 0.84 and VAF = 78.16 for the FF condition, and R-squared = 0.74 and
VAF = 78.04 for the HFS-FF condition. These metrics indicate that our model provides an adequate
representation of the observed learning behavior. We added this point to the manuscript (lines 646-
653) and in Supplemental Figure S8.

**Figure R8: Condition-specific goodness of fit.** Calculated retention factor and error sensitivity were used to
 compare the model to the data (median of trial-dependent error calculated for the learning sessions) for testing
 the R-squared and variance accounted for in the FF (left) and FF-HFS (right) conditions.

3. Dimensionality. Determining dimensionality is always tricky in real data, and different measures
 tell you different things. Here, the authors use the Participation Ratio, which works purely on the
 eigenvectors – the distribution of variance across dimensions. It does not (cannot) distinguish “real”
 variance from noise. Adding a real, high-dimensional signal pushes the PR up, but so does adding
 noise, which is also generally high-dimensional. Adding a low-dimensional signal, either a large one
 or one that is redundant with information already in the system, pushes the PR down given the exact
 same noise level. This makes it hard to interpret Figure 5a,c. The authors argue that cerebellar inputs
 “quench the dimensionality of the cortical manifold”. Alternatively, the small drop observed in
 dimensionality could be due to a reduction in noise at cue onset (see papers by Churchland and
 Shenoy, Rajan and Abbott, and others); or they could be due to the addition of a real low-D signal
 from any of a number of inputs, including motor thalamus, on which there is a burgeoning literature.
 The higher dimensionality during HFS could be due to the loss of a low-D input as the authors
 suggest, or could be due to the injection of noise, or to reduction of stability in the network for any
 reason (which we might expect from the increased behavioral variability). Looking at Supplementary
 Figure 3, the drop in estimated dimensionality seems to occur just after Cue and reach a minimum
 just before Go. This is a similar time course to the quenching of variability known in premotor cortex
 (e.g., (Churchland, Yu et al. 2010)). In sum, it is hard to know which of several sources causes the
 effects on dimensionality observed here, and interpretation should be cautious.

We greatly appreciate the reviewer’s thoughtful analysis of the challenges in interpreting
 dimensionality in neural data, and in particular the role of noise in increasing the participation ratio
 (PR). The reviewer correctly highlights that high-dimensional noise can raise the effective
 dimensionality. However, we would like to clarify several points and report additional analyses to
 support our interpretation that the observed increase in dimensionality reflects a change in the
 underlying signal rather than an increase in noise.

**No Increase in Trial-to-Trial Noise.** We acknowledge the reviewer’s concern that noise could
 contribute to the observed increase in dimensionality during the cerebellar block. While we reported
 in the manuscript that there was no significant change in trial-to-trial variability, as assessed by the

Fano Factor (FF) of single-neuron activity, this point may not have been emphasized sufficiently. We
 appreciate the opportunity to elaborate further here.

To address this concern in more detail, we calculated the FF of the single-neuron averaged activity in
 200 ms windows. Our analysis showed no significant change in the FF at any time point in the trial,
 including the pre-go window where the effective dimensionality was calculated. Specifically, we
 compared the FF distributions across neurons between the control and cerebellar block conditions
 and observed no significant differences. This can be seen in **Figure R9** below: Panels A and B show
 the pairwise comparison of the Fano factors for the Control and HFS (for single cells). The p-value of
 the paired t-test appears in the legend for each panel. Panel C shows the distribution of single-cell
 Fano factors in a window between -300 and +100 ms around the Go signal shown after subtracting
 the baseline level for the cell and Panel D shows the same as C but without the baseline subtraction.
 This finding suggests that the trial-to-trial variability remained stable, making it unlikely that the
 observed increase in dimensionality was driven by noise.

**Figure R9: Comparing the Fano Factor in control and HFS trials.** The pre-go Fano Factor was calculated
 around the Go signal (-300 ms to + 100 ms) when the monkeys made repeated movements to the Learned
 Target. **A**, pairwise comparison of Fano factors obtained for single cells in the control (x-axis) and HFS (y-axis)
 conditions. **B**, Same as **A** but for FF (x-axis) and FF-HFS (y-axis) conditions. In both cases a paired t-test
 revealed no significant differences. **C**, Population-based comparison of mean changes in Fano factor
 (reduction in the Fano factor during the pre-Go vs. pre-Cue levels) calculated during HFS and Control
 conditions. **D**, same as **C** but comparing the non-subtracted Fano factor. No significant differences were found
 in either case.

**Noise-Robust Dimensionality Estimate.** Our analysis focused on target-related activity averaged
 across multiple trials rather than single-trial data. This approach is essential when pooling data

across different recording sessions. Consequently, the dimensionality analysis was inherently more
 robust to noise, as target-averaged data diminish the impact of trial-to-trial fluctuations that could
 inflate dimensionality estimates in single-trial analyses, primarily by flattening the spectrum at
 higher eigenvalues of the correlation matrix. Conversely, in target-averaged data, the effects of noise
 result in small fluctuations of the mean activity vector due to finite trial samples, and the impact of
 uncorrelated noise on dimensionality is significantly attenuated. Thus, our dimensionality calculation
 reflects the aggregate structure of the signal rather than the noise.

To further strengthen our point, we tested how much single-trial noise is required to increase the
 dimensionality such that it would replicate the observed increase in dimensionality. To this end, we
 first subsampled a number of control trials to match the number of HFS trials. Then, for each trial,
 we added a random noise term $a \xi_i^n$ where ξ_i^n was a random number sampled, i.i.d. from a normal
 distribution with zero mean and variance σ_n^2 , which was the calculated variance of the n neuron in
 all control trials, i was a trial index and a was a parameter which we varied to mimic different levels
 of noise. We repeated the analysis many times for several values of a . The results, as depicted in
 **Figure R10** below, show that to make the dimensionality of the control data insignificantly different
 from HFS, we would have to multiply the single-trial noise variance by a factor of roughly 2 (bars
 show the standard deviation of the added noise and the error bars indicate the standard deviation
 with $n=100$). This analysis further supports our claim that the dimensionality increase under HFS
 cannot be accounted for by noise.

**Figure R10: Effect of noise on estimated D_{pca} .** Calculating the effect of artificially injecting noise to the
 control data on mean D_{pca} .

**Decrease in FF before movement is not disturbed by the cerebellar block.** We appreciate the
 reviewer's comment on the similarity between the temporal dynamics of dimensionality changes
 observed in our data and the quenching of variability in motor regions reported in previous studies.
 We indeed saw a decrease in the FF before movement onset, which aligns with the results reported
 in (Churchland, Yu et al. 2010). However, the decrease in FF was similar in both the Control and HFS
 conditions. Furthermore, there were no significant differences between the FF for the Control and
 HFS, as stated above. We thus concluded that the decrease in FF was not of cerebellar origin and did
 not explain the change in the cortical dimensionality that we observed. Finally, quenching variability
 does not necessarily imply a dimensionality reduction, and vice-versa. Our primary focus was on the
 differences in underlying dimensionality between the Control conditions and the cerebellar block
 rather than changes across different time points.

**Conclusion.** Given that the effects of noise on our analysis are small—owing to both the absence of
increased trial-to-trial variability in the data and the use of target-averaged activity—we suggest that
the underlying signal (and not noise) was the primary driver of the observed changes in
dimensionality. Specifically, we interpret the increased dimensionality during the cerebellar block as
reflecting the loss of a low-dimensional input signal, rather than an increase in noise or instability.
This interpretation is consistent with our observations and with the expected role of cerebellar
inputs in structuring the cortical activity manifold.

We now explicitly state in the main text that the increase in dimensionality was not due to an
increase in the trial-to-trial noise (line 282-284), and we have added the figures showing the Fano
factor to the supplementary material (Supplemental Figure S6).

4. Model. I don't have any specific problem with the model in Figure 5, but I was unclear why it took
this structure in particular. If the motivation was solely from the dimensionality results, I explain
above why this is not enough to motivate a single model. If there's literature motivating it, please
cite it. If it's just a plausibility argument, please make this clearer.

We appreciate the reviewer's comment and would like to clarify the motivation behind our model.
As mentioned earlier, our data suggest that the observed changes in dimensionality were driven by
changes in the underlying signal rather than alterations in noise or trial-to-trial variability. The model
aims to capture how removing a low-dimensional input from a recurrent cortical network can lead to
an increase in dimensionality and a reduction in generalization performance, consistent with our
experimental observations during the cerebellar block.

We chose this particular model structure because it represents the simplest architecture that
captures the key dynamics of cerebellar influence on cortical activity. Specifically, it demonstrates
how cerebellar input, which is typically low-dimensional and structured, can constrain cortical
dimensionality and support task performances.

Similar models have been used in previous work to study recurrent network dynamics and
representations (e.g. (Sussillo and Abbott 2009, Mastrogiuseppe and Ostojic 2018, Kadmon,
Timcheck et al. 2020)). Simplified models of cortical-cerebellar interactions have been explored in
recent studies further supporting the plausibility of our approach. This model structure has also
been used as a simple framework for studying motor planning in the cortex (e.g., (Kao, Sadabadi et
al. 2021, Schimel, Kao et al. 2024)). We have added these references to the manuscript (lines 786-
788) to reinforce the relevance and validity of our model choice.

While the model is not the only possible structure, it serves as a parsimonious framework to
illustrate how cerebellar inputs affect cortical representations. Alternative models incorporating
both low-dimensional and high-dimensional inputs could yield similar results, but we opted for this
straightforward approach for clarity and to directly connect our experimental findings to theoretical
predictions.

Importantly, the model was intended to be a plausible demonstration rather than an exhaustive
account illustrating that the loss of a low-dimensional cerebellar input can explain the observed
increase in cortical dimensionality. This point has been added to the manuscript, in line 285.

Other

5. The second control for higher trial-by-trial error due to HFS, of categorizing sessions as low or high

noise, is unhelpful. The primary control, in which the distributions are matched, is fine. It's not clear
 to me why the second control would help in any way, since even within noise group you're still
 comparing higher noise in the FF-HFS condition to lower noise in the FF.

The reviewer is right and the point we were trying to make is perhaps slightly redundant, although
 we thought it complements our point showing that the effect of HFS on the noise level cannot fully
 explain the concomitant reduction in error sensitivity. Specifically, we added this analysis to verify
 that using sessions with a matched noise level did not mask any possible effect of noise magnitude.
 For instance, consider a scenario where the error sensitivity depends solely on the noise magnitude.
 In this case, the control for matching noise level could mask this effect since we consider all matched
 sessions to be independent of the underlying noise level (as long as the level is comparable between
 the two conditions). We therefore divided sessions solely based on their noise level, independently
 for the Control and HFS (which means that the two distributions of low (or high) noise levels shown
 in the left (right) panel of Figure 2e depict data collected from different (non-matching) Control and
 HFS sessions.

6. Figure 1h: the FF regression line does not appear to fit the data well at all. This may be the best-fit
 line because of the huge distribution for that first point (i.e., I'm not suggesting this is an outright
 error), but that fit is far from convincing. Is it just this way of visualizing the data, or is there a
 problem here? Can the authors justify this model when it fits so poorly?

The reviewer is right (see also point 7.2 by Reviewer #1) and the figure in the paper presented the
 relationship between the adaptive response and error for only one monkey. Studying the results for
 each of the two monkeys (Figure R2,A-C, shown again here for convenience) reveals that the same
 effect was replicated (at different strengths) for the two monkeys and was also visible when
 combining the data from the two monkeys (Figure R2D). The results were not identical, but the
 trend was similar and the significance tests revealed similar results for both monkeys.

**Figure R2: Relations between adaptive response and error in different trials conditions.** Monkey-specific
 adaptive responses were calculated for each monkey during the Control conditions (A), HFS (B) and FF (C).
 The regression slopes are shown for each graph. D Mean adaptive response calculating by pooling data from
 both monkeys. Each r value was tested for a significance difference from zero slope (t -test). Differences in
 regression slopes were tested and found to be non-significant when comparing the Control vs. HFS conditions,
 but statistically significant when comparing the Control vs. FF conditions.

Please note that the purpose of this analysis was to show that HFS in itself does not act as an
 external source of noise which might reduce the error sensitivity. This is important since it lends
 weight to our claim that the impaired learning under HFS is a fundamental property, similar to the

701 observed deficits in cerebellar patients, and not a simple outcome of increased motor noise.
 However, the reviewer is right and even through the correlation between the adaptive response and
 the error was significant, it is not very strong. We hope that the similarity of results between the two
 monkeys provides a more convincing support for our claim. The figure in the manuscript now
 includes the combined data for the two monkeys and we added a supplemental figure
 (Supplemental Figure S1) which contains single monkey data. The text was modified accordingly
 (lines 170-175)

7. Interpretation of the PCA plots (Figure 3)

7a. PC2 has a large offset between control and HFS, which appears to show that the HFS is having a
 direct effect on neural firing rates. This is presumably from the tonic activation of the cerebello-
 cortical pathway. Given that it pushes firing rates away from their typical values, HFS may perturb
 the circuit's function directly and not solely through preventing normal phasic and structured inputs.
 This should be mentioned.

We thank the reviewer for this comment. This is true and we think that part of the effect of HFS is a
 tonic drive to the motor cortex. This point has been added to the manuscript, line 204-205.

7b. The interpretation of Figure 3h is suboptimal. The authors describe the effect of HFS as “inverse”
 vs. in other PCs. This is literally true, but somewhat misleading. The larger picture is that the pattern
 of targets is slightly bent in 5-D, and although you can find a dimension in which representation is
 flipped, it probably doesn't give a good picture of the larger structure.

The reviewer is right and the explanation for Figure 3h was suboptimal. In this figure we present the
 target-dependent effects of HFS on neural states compared to the Control conditions. Specifically,
 PC5 differentiates between the Control and HFS for both distal targets located away from the body
 (blue-shaded targets in Fig. 3h) vs. proximal targets located near the body (red-shaded target).
 However, this dependency was reversed between the two groups of targets (positive/negative
 values for control/HFS in distal targets and the reverse for proximal targets). The point we were
 trying to make is that the cerebellar input contains **target-related information** such that its
 contribution to the cortical representation is not simply a uniform tonic drive (as might be deduced
 from Fig. 3C). This part of the text was modified accordingly (line 204-207).

8. Please choose a different name for your dimensionality measure. “dPCA” is already the name of a
 method. “D” would be fine.

Change was made.

Minor editing error and typos

Supp Fig 2: is the on-figure legend wrong here? Colors in the on-figure legend are reversed relative
 to the caption and to Fig 4.

The reviewer is right and the legends on the figure were reversed. We rectified this mistake.

L190: citations not inserted correctly

Citation was rectified

L231: typo: should be TDR, not TRD

"tr_thresh" written "tr_tresh" in several places

Typos were corrected

**References**

- Albert, S. T., J. Jang, H. R. Sheahan, L. Teunissen, K. Vandevoorde, D. J. Herzfeld and R. Shadmehr
 (2021). "An implicit memory of errors limits human sensorimotor adaptation." Nat Hum Behav
 5(7): 920-934.
- Churchland, M. M., B. M. Yu, J. P. Cunningham, L. P. Sugrue, M. R. Cohen, G. S. Corrado, W. T.
 Newsome, A. M. Clark, P. Hosseini, B. B. Scott, D. C. Bradley, M. A. Smith, A. Kohn, J. A.
 Movshon, K. M. Armstrong, T. Moore, S. W. Chang, L. H. Snyder, S. G. Lisberger, N. J. Priebe, I.
 755 M. Finn, D. Ferster, S. I. Ryu, G. Santhanam, M. Sahani and K. V. Shenoy (2010). "Stimulus onset
 quenches neural variability: a widespread cortical phenomenon." Nat Neurosci 13(3): 369-378.
- Hore, J. and D. Flament (1988). "Changes in motor cortex neural discharge associated with the
 development of cerebellar limb ataxia." J Neurophysiol 60(4): 1285-1302.
- Kadmon, J., J. Timcheck and S. Ganguli (2020). "Predictive coding in balanced neural networks with
 noise, chaos and delays." Advances in neural information processing systems 33: 16677-16688.
- Kao, T. C., M. S. Sadabadi and G. Hennequin (2021). "Optimal anticipatory control as a theory of
 motor preparation: A thalamo-cortical circuit model." Neuron 109(9): 1567-1581 e1512.
- Mastrogiuseppe, F. and S. Ostojic (2018). "Linking Connectivity, Dynamics, and Computations in Low-
 Rank Recurrent Neural Networks." Neuron 99(3): 609-623 e629.
- Nashef, A., O. Cohen, R. Harel, Z. Israel and Y. Prut (2019). "Reversible Block of Cerebellar Outflow
 Reveals Cortical Circuitry for Motor Coordination." Cell Rep 27(9): 2608-2619 e2604.
- Paz, R., T. Boraud, C. Natan, H. Bergman and E. Vaadia (2003). "Preparatory activity in motor cortex
 reflects learning of local visuomotor skills." Nat Neurosci 6(8): 882-890.
- Schimel, M., T. C. Kao and G. Hennequin (2024). "When and why does motor preparation arise in
 recurrent neural network models of motor control?" Elife 12.
- Shenoy, K. V., M. T. Kaufman, M. Sahani and M. M. Churchland (2011). "A dynamical systems view of
 motor preparation: implications for neural prosthetic system design." Prog Brain Res 192: 33-
 58.
- Sun, X., D. J. O'Shea, M. D. Golub, E. M. Trautmann, S. Vyas, S. I. Ryu and K. V. Shenoy (2022).
 "Cortical preparatory activity indexes learned motor memories." Nature 602(7896): 274-279.
- Sussillo, D. and L. F. Abbott (2009). "Generating coherent patterns of activity from chaotic neural
 networks." Neuron 63(4): 544-557.
- Vyas, S., D. J. O'Shea, S. I. Ryu and K. V. Shenoy (2020). "Causal Role of Motor Preparation during
 Error-Driven Learning." Neuron 106(2): 329-339 e324.
